# SIMONe: View-Invariant, Temporally-Abstracted Object Representations via Unsupervised Video Decomposition

**Rishabh Kabra**[1], **Daniel Zoran**[1], **Goker Erdogan**[1], **Loic Matthey**[1]
**Antonia Creswell**[1], **Matthew Botvinick**[1], **Alexander Lerchner**[1], **Christopher P. Burgess**[2*]
[1]DeepMind, [2]Wayve, [*]Work done at DeepMind
{rkabra, danielzoran, gokererdogan, lmatthey,
tonicreswell, botvinick, lerchner}@deepmind.com, chrisburgess@wayve.ai

## Abstract

To help agents reason about scenes in terms of their building blocks, we wish to extract the compositional structure of any given scene (in particular, the configuration and characteristics of objects comprising the scene). This problem is especially difficult when scene structure needs to be inferred while also estimating the agent's location/viewpoint, as the two variables jointly give rise to the agent's observations. We present an unsupervised variational approach to this problem. Leveraging the shared structure that exists across different scenes, our model learns to infer two sets of latent representations from RGB video input: a set of "object" latents, corresponding to the time-invariant, object-level contents of the scene, as well as a set of "frame" latents, corresponding to global time-varying elements such as viewpoint. This factorization of latents allows our model, SIMONe, to represent object attributes in an allocentric manner which does not depend on viewpoint. Moreover, it allows us to disentangle object dynamics and summarize their trajectories as time-abstracted, view-invariant, per-object properties. We demonstrate these capabilities, as well as the model's performance in terms of view synthesis and instance segmentation, across three procedurally generated video datasets.

## 1 Introduction

The problem of *unsupervised visual scene understanding* has become an increasingly central topic in machine learning [1, 2]. The attention is merited by potential gains to reasoning, autonomous navigation, and myriad tasks. However, within the current literature, different studies frame the problem in different ways. One approach aims to decompose images into component objects and object features, supporting (among other things) generation of alternative data that permits insertion, deletion, or repositioning of individual objects [3–6]. Another approach aims at a very different form of decomposition—between allocentric scene structure and a variable viewpoint—supporting generation of views of a scene from new vantage points [7–9] and, if not supplied as input, estimation of camera pose [10]. Although there is work pursuing both of these approaches concurrently in the supervised setting [11–13], very few previous studies have approached the combined challenge in the unsupervised case. In this work, we introduce the Sequence-Integrating Multi-Object Net (SIMONe), a model which pursues that goal of object-level and viewpoint-level scene decomposition and synthesis without supervision. SIMONe is designed to handle these challenges without privileged information concerning camera pose, and in dynamic scenes.

Given a video of a scene our model is able to decouple scene structure from viewpoint information (see Figure 1). To do so, it utilizes video-based cues, and a structured latent space which separates

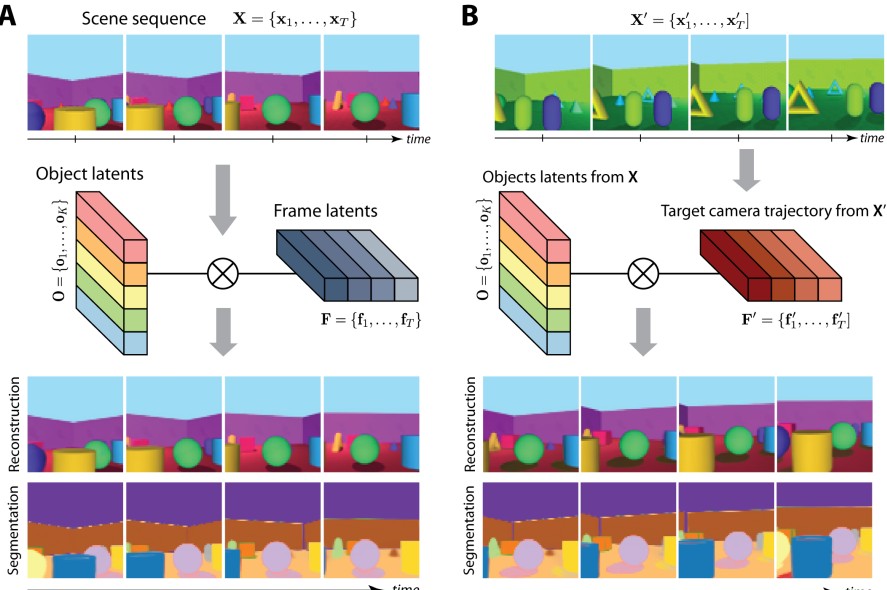

Figure 1: **Decomposition (A):** SIMONe factorizes a scene sequence $\mathbf{X}$ into *scene content* ("object latents," constant across the sequence) and *view/global content* ("frame latents," one per frame) without supervision. Its spatio-temporal attention-based inference naturally allows stable object tracking (e.g. the green sphere is assigned the same segment across frames). **Recomposition (B):** Object latents of a given sequence $\mathbf{X}$ can be recomposed with the frame latents of a different (i.i.d.) sequence $\mathbf{X}'$ to generate a consistent rendering of the same scene (i.e. objects and their properties, relative arrangements, and segmentation assignments) from entirely different viewpoints. Notice that both camera pose and lighting are transferred, as evidenced by the wall corners in the background and the shadows of the green sphere.

time-invariant per-object features from time-varying global features. These features are inferred using a transformer-based network which integrates information jointly across space and time.

Second, our method seeks to summarize objects' dynamics. It learns to disentangle not only static object attributes (and their 2D spatial masks), but also object trajectories, without any prior notion of these objects, from videos alone. The learnt trajectory features are temporally abstract and per object; they are captured independently of the dynamics of camera pose, which being a global property, is captured in the model's per-frame (time-varying) latents.[1]

Our model thus advances the state of the art in unsupervised, object-centric scene understanding by satisfying the following desiderata: **(1)** decomposition of multi-object scenes from RGB videos alone; **(2)** handling of changing camera pose, and simultaneous inference of scene contents and viewpoint from correlated views (i.e. sequential observations of a moving agent); **(3)** learning of structure across diverse scene instances (i.e. procedurally sampled contents); **(4)** object representations which summarize static object attributes like color or shape, view-dissociated properties like position or size, as well as time-abstracted trajectory features like direction of motion; **(5)** no explicit assumptions of 3D geometry, no explicit dynamics model, no specialized renderer, and few a priori modeling assumptions about the objects being studied; and **(6)** simple, scalable modules (for inference and rendering) to enable large-scale use.

## 2 Related Work

Given the multifaceted problem it tackles, SIMONe connects across several areas of prior work. We describe its nearest neighbors from three scene understanding domains below:

---

[1]Animated figures are at https://sites.google.com/view/simone-scene-understanding/.

**Scene decomposition models.** (**1**) Our work builds on a recent surge of interest in unsupervised scene decomposition and understanding, especially using slot structure to capture the objects in a scene [14]. One line of work closely related to SIMONe includes methods like [3–6, 15], which all share SIMONe's Gaussian mixture pixel likelihood model. While these prior methods handled only static scenes, more recent work [16–18, 12] has extended them to videos with promising results. Nevertheless, these approaches have no mechanism or inductive bias to separate view information from scene contents. Moreover, many of them are conditioned on extra inputs like the actions of an agent/camera to simplify inference. (**2**) Another family of decomposition models originated with Attend, Infer, Repeat (AIR) [19]. AIR's recurrent attention mechanism does split images into components with separate appearance and pose latents each. Later work [6, 20–23] also extended the model to videos. Despite their structured latents, these models do not learn to distill object appearance into a time-invariant representation (as their appearance and pose latents are free to vary as a function of time). They also require separate object discovery and propagation modules to handle appearing/disappearing objects. In contrast, SIMONe processes a full sequence of images using spatio-temporal attention and produces a single time-invariant latent for each object, hence requiring no explicit transition model or discovery/propagation modules.

**Multi-view scene rendering models.** Models which assume viewpoint information for each image like GQN [7], SRNs [8], and NeRF [9] have shown impressive success at learning implicit scene representations and generating novel views from different viewpoints. Recent work [24–27] has further extended these models to videos using deformation fields to model changes in scene geometry over time. In contrast to SIMONe, these models can achieve photorealistic reconstructions by assuming camera parameters (viewpoint information). To allow a direct comparison, we use a view-supervised version of SIMONe in Section 4.1. There is also recent work [28, 29] that relaxes the known viewpoint constraint, but they still model single scenes at a time, which prevents them from exploiting regularities over multiple scenes. A more recent line of work [30–32] explored amortizing inference by mapping from a given set of images to scene latents, but they cannot handle videos yet. Note that all of these models treat the whole scene as a single entity and avoid decomposing it into objects. One exception here is [33], which represents objects with separate pose and appearance latents. However, this model is purely generative and cannot infer object latents from a given scene. Another exception is [13], which can in fact infer object representations, but nevertheless depends on view supervision.

**Simultaneous localization and mapping.** The problem of inferring scene representations in a novel environment by exploring it (rather than assuming given views and viewpoint information) is well studied in robotics and vision [10]. Classic SLAM techniques often rely on EM [34, 35] or particle filters [36] to infer viewpoint and scene contents jointly. While our problem is slightly simpler (we can leverage shared structure across scene instances; certain elements such as the shape of the room are held constant; and we use offline data rather than active exploration), our approach of using a factorized variational posterior provides a learning-based solution to the same computational problem. Our simplified setting is perhaps justified by our unsupervised take on the problem. On the other hand, we don't assume simplifications which may be common in robotics practice (e.g. known camera properties like field of view; or the use of multiple cameras or depth sensors). Most popular SLAM benchmarks [37–39] are on unstructured 3D scenes and hence it was not straightforward for us to compare directly to classic methods. But an encouraging point of overlap is that object-centric SLAM formulations [11] as well as learning-based solutions [40, 41] are active topics of research. Our work could open new avenues in object-centric scene mapping without supervision.

## 3   Model

SIMONe is a variational auto-encoder [42] consisting of an inference network (encoder) which infers latent variables from a given input sequence, and a generative process (decoder) which decodes these latents back into pixels. Using the Evidence Lower Bound (ELBO), the model is trained to minimize a pixel reconstruction loss and latent compression KL loss. Crucially, SIMONe relies on a factorized latent space which enforces a separation of static object attributes from global, dynamic properties such as camera pose. We introduce our latent factorization and generative process in Section 3.1. Then in Section 3.2, we describe how the latents can be inferred using a transformer-based encoder, significantly simplifying the (recurrent or autoregressive) architectures used in prior work. Finally, we fully specify the training scheme in Section 3.3.

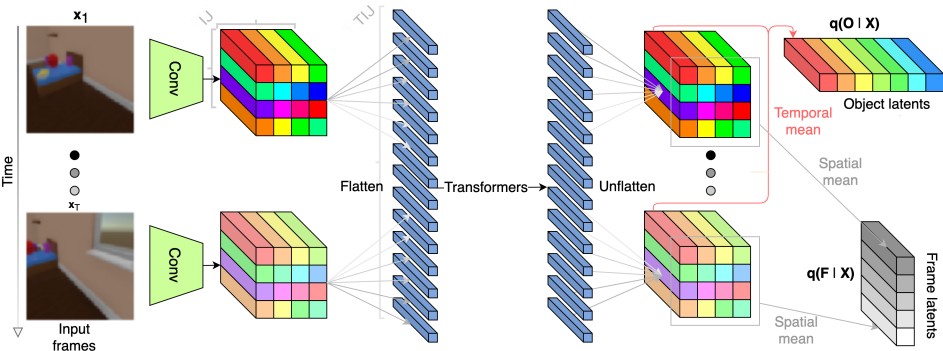

Figure 2: **Architecture** of the SIMONe inference network $\mathcal{E}_\phi$. The transformers integrate information jointly across space and time to infer (the posterior parameters of) the object and frame latents.

## 3.1 Latent Structure and Generative Process

Our model aims to capture the structure of a scene, observed as a sequence of images from multiple viewpoints (often along a smooth camera trajectory, though this is not a requirement). Like many recently proposed object-centric models we choose to represent the scene as a set of $K$ *object* latent variables $\mathbf{O} := \{\mathbf{o}_k\}_{k=1}^K$. These are invariant by construction across all frames in the sequence (i.e. their distribution is constant through time, and expected to summarize information across the whole sequence). We also introduce $T$ *frame* latents $\mathbf{F} := \{\mathbf{f}_t\}_{t=1}^T$, one for each frame in the sequence, that capture time-varying information. Note that by choosing this factorization we reduce the number of required latent variables from $K \cdot T$ to $K + T$. The latent prior $p(\mathbf{O}, \mathbf{F}) = \prod_k \mathcal{N}(\mathbf{o}_k \mid \mathbf{0}, \mathbf{I}) \prod_t \mathcal{N}(\mathbf{f}_t \mid \mathbf{0}, \mathbf{I})$ is a unit spherical Gaussian, assuming and enforcing independence between object latents, frame latents, and their feature dimensions.

Given the latent variables, we assume all pixels and all frames to be independent. Each pixel is modeled as a Gaussian mixture with $K$ components. The mixture weights for pixel $\mathbf{x}_{t,i}$ capture which component $k$ "explains" that pixel ($1 \leq i \leq HW$). The mixture logits $\hat{m}_{k,t,i}$ and RGB (reconstruction) means $\boldsymbol{\mu}_{k,t,i}$ are computed for every component $k$ at a specific time-step $t$ and specific pixel location $\mathbf{l}_i$ using a decoder $\mathcal{D}_\theta$:

$$\hat{m}_{k,t,i}, \boldsymbol{\mu}_{k,t,i} = \mathcal{D}_\theta(\mathbf{o}_k, \mathbf{f}_t; \mathbf{l}_i, t) \tag{1}$$

$$p(\mathbf{x}_{t,i} \mid \mathbf{o}_1, ..., \mathbf{o}_K, \mathbf{f}_t; t, \mathbf{l}_i) = \sum_k m_{k,t,i} \mathcal{N}(\mathbf{x}_{t,i} \mid \boldsymbol{\mu}_{k,t,i}; \sigma_x) \tag{2}$$

We decode each pixel independently, "querying" our *pixel-wise* decoder using the sampled latents, coordinates $\mathbf{l}_i \in [-1, 1]^2$ of the pixel, and time-step $t \in [0, 1)$ being decoded as inputs. The decoder's architecture is an MLP or 1x1 CNN. (See Appendix A.3.1 for the exact parameterization as well as a diagram of the generative process). By constraining the decoder to work on individual pixels, we can use a subset of pixels as training targets (as opposed to full images; this is elaborated in Section 3.3). Once they are decoded, we obtain the mixture weights $m_{k,t,i}$ by taking the softmax of the logits across the $K$ components: $m_{k,t,i} = \texttt{softmax}_k(\hat{m}_{k,t,i})$. Equation 2 specifies the full pixel likelihood, where $\sigma_x$ is a scalar hyperparameter.

## 3.2 Inference

Given a sequence of frames $\mathbf{X} := \{\mathbf{x}_t\}_{t=1}^T$ we now wish to infer the corresponding object latents $\mathbf{O}$ and frame latents $\mathbf{F}$. The exact posterior distribution $p(\mathbf{O}, \mathbf{F} \mid \mathbf{X})$ is intractable so we resort to using a Gaussian approximate posterior $q(\mathbf{O}, \mathbf{F} \mid \mathbf{X})$. The approximate posterior is parameterized as the output of an inference (encoder) network $\mathcal{E}_\phi(\mathbf{X})$ which outputs the mean and (diagonal) log scale for all latent variables given the input sequence.

SIMONe's inference network is based on the principle that spatio-temporal data can be processed *jointly* across space and time using transformers. Beyond an initial step, we don't need the translation invariance of a CNN, which forces spatial features to interact gradually via a widening receptive field. Nor do we need the temporal invariance of an RNN which forces sequential processing. Instead, we

let feature maps interact simultaneously across the cross-product of space and time. See Figure 2 for an overview of our encoder architecture implementing this.

Concretely, each frame $\mathbf{x}_t$ in the sequence is passed through a CNN which outputs $IJ$ spatial feature maps at each time-step (containing $C$ channels each). $IJ$ can be larger than the number of object latents $K$. (For all results in the paper, we set $I$ and $J$ to 8 each, and $K = 16$.) The rest of the inference network consists of two transformers $\mathcal{T}_1$ and $\mathcal{T}_2$. $\mathcal{T}_1$ takes in all $TIJ$ feature maps. Each feature map attends to all others as described. $\mathcal{T}_1$ outputs $TIJ$ transformed feature maps. When $IJ > K$, we apply a spatial pool to reduce the number of slots to $TK$ (see Appendix A.3.2 for details). These slots serve as the input to $\mathcal{T}_2$, which produces an equal number of output slots. Both transformers use absolute (rather than relative) positional embeddings, but these are 3D to denote the spatio-temporal position of each slot. We denote the output of $\mathcal{T}_2$ as $\hat{\mathbf{e}}_{k,t}$. This intermediate output is aggregated along separate axes (and passed through MLPs) to obtain $T$ frame and $K$ object posterior parameters respectively. Specifically, $\boldsymbol{\lambda}_{\mathbf{o}_k} = \mathrm{mlp}_o(1/T \sum_t \hat{\mathbf{e}}_{k,t})$ while $\boldsymbol{\lambda}_{\mathbf{f}_t} = \mathrm{mlp}_f(1/K \sum_k \hat{\mathbf{e}}_{k,t})$. Using these posterior parameters we can sample the object latents $\mathbf{o}_k \sim \mathcal{N}(\boldsymbol{\lambda}_{\mathbf{o}_k}^{\mu}, \exp(\boldsymbol{\lambda}_{\mathbf{o}_k}^{\sigma})\mathbb{1})$, and the frame latents $\mathbf{f}_t \sim \mathcal{N}(\boldsymbol{\lambda}_{\mathbf{f}_t}^{\mu}, \exp(\boldsymbol{\lambda}_{\mathbf{f}_t}^{\sigma})\mathbb{1})$.

### 3.3 Loss and Training

The model is trained end to end by minimizing the following negative-ELBO derivative:

$$
\frac{-\alpha}{T_d H_d W_d} \sum_{t=1}^{T_d} \sum_{i=1}^{H_d W_d} \log p(\mathbf{x}_{t,i} \mid \mathbf{o}_1, ..., \mathbf{o}_K, \mathbf{f}_t; t, \mathbf{l}_i) + \frac{\beta_o}{K} \sum_k D_{KL}(q(\mathbf{o}_k \mid \mathbf{X}) \,\|\, p(\mathbf{o}_k))
$$
$$
+ \frac{\beta_f}{T} \sum_t D_{KL}(q(\mathbf{f}_t \mid \mathbf{X}) \,\|\, p(\mathbf{f}_t))
$$

(3)

We normalize the data log-likelihood by the number of decoded pixels $(T_d H_d W_d)$ to allow for decoding fewer than all input pixels $(THW)$. This helps scale the size of the decoder (without reducing the learning signal, due to the correlations prevalent between adjacent pixels). Normalizing by $1/T_d H_d W_d$ ensures consistent learning dynamics regardless of the choice of how many pixels are decoded. $\alpha$ is generally set to 1, but available to tweak in case the scale of $\beta_o$ and $\beta_f$ is too small to be numerically stable. Unless explicitly mentioned, we set $\beta_o = \beta_f$. See Appendix A.3 for details.

## 4 Comparative Evaluation

To evaluate the model we focus on two tasks: novel view synthesis and video instance segmentation. On the first task (Section 4.1), we highlight the benefit of view information when it is provided as ground truth to a simplified version of our model (denoted "SIMONe-VS" for view supervised), as well as baseline models like GQN [7] and NeRF-VAE [43]. On the second task (Section 4.2), we deploy the fully unsupervised version of our model; we showcase not only the possibility of inferring viewpoint from data, but also its benefit to extracting object-level structure in comparison to methods like MONet [3], Slot Attention [15], and Sequential IODINE [4].

Our results are based on three procedurally generated video datasets of multi-object scenes. In increasing order of difficulty, they are: **Objects Room 9** [44], **CATER** (moving camera) [45], and **Playroom** [46]. These were chosen to meet a number of criteria: we wanted at least 9-10 objects per scene (there can be fewer in view, or as many as 30 in the case of Playroom). We wanted a moving camera with a randomized initial position (the only exception is CATER, where the camera moves rapidly but is initialized at a fixed position to help localization). We wanted ground-truth object masks to evaluate our results quantitatively. We also wanted richness in terms of lighting, texture, object attributes, and other procedurally sampled elements. Finally, we wanted independently moving objects in one dataset (to evaluate trajectory disentangling and temporal abstraction), and unpredictable camera trajectories in another (the Playroom dataset is sampled using an arbitrary agent policy, so the agent is not always moving). Details on all datasets are in Appendix A.2.

### 4.1 View synthesis (with viewpoint supervision)

We first motivate view-invariant object representations by considering the case when ground-truth camera pose is provided to our model (a simplified variant we call "SIMONe-VS"). In this scenario,

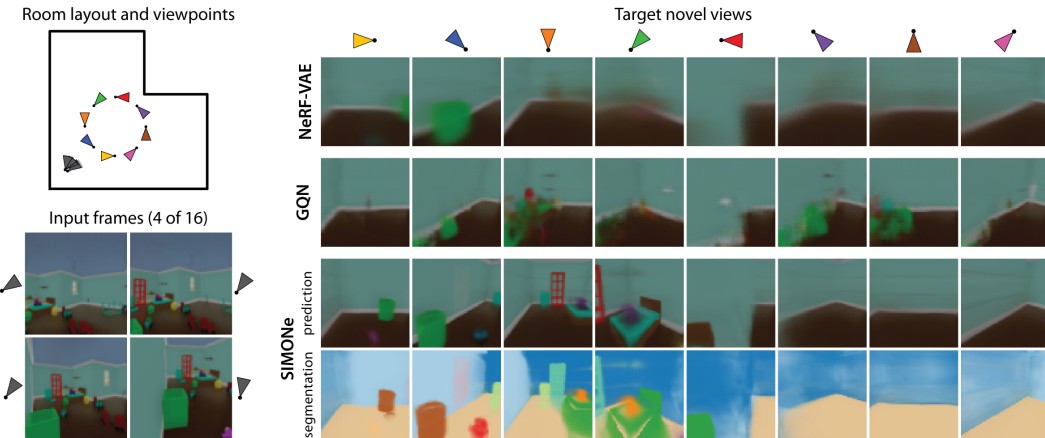

Figure 3: **Comparison of scene representation and view synthesis capabilities** between SIMONe-VS, NeRF-VAE, and GQN. All models partially observe a procedurally generated Playroom from a given sequence of frames (we visualize 4 of the 16 input frames fed to the models). Then, we decode novel views on a circular trajectory around the room, with the yaw linearly spaced in $[-\pi, \pi]$. NeRF-VAE retains very little object structure, while GQN hallucinates content. SIMONe-VS can produce fine reconstructions of objects that it observes even partially or at a distance (such as the bed or shelves in the scene). SIMONe-VS also segments the scene as a bonus. See Appendix A.5.1 for similar plots from different scenes/input sequences.

we don't infer any frame latents. Rather, the encoder and decoder are conditioned on the viewpoint directly. This *view-supervised* setting is similar to models like GQN and NeRF which represent the contents of a scene implicitly and can be queried in different directions.

We compare three such models on view synthesis in the Playroom. The models are provided a set of 16 consecutive frames as context, partially revealing a generated room. Having inferred a scene representation from the input context, the models are then tasked with generating unobserved views of the scene. This extrapolation task is performed without any retraining, and tests the coherence of the models' inferred representations. The task is challenging given the compositional structure of each Playroom scene, as well as the variation across scenes (the color, position, size, and choice of all objects are procedurally sampled per scene; only the L-shaped layout of the room is shared across scenes in the dataset). Because each model is trained on and learns to represent many Playroom instances, NeRF itself is not directly suitable for the task. It needs to be retrained on each scene, whereas we want to infer the specifics of any given room at evaluation time. NeRF-VAE addresses this issue and makes it directly comparable to our model.

To set up the comparison, we first trained SIMONe-VS and evaluated its log-likelihood on Playroom sequences. Then, we trained GQN and NeRF-VAE using constrained optimization (GECO [47]) to achieve roughly the same log likelihood per pixel. See Appendix A.5.1 for a comparison of the models in terms of the reconstruction-compression trade-off.

Qualitatively, the models show vast differences in their perceived structure (see Figure 3). NeRF-VAE blurs out nearly all objects in the scene but understands the geometry of the room and is able to infer wall color. GQN produces more detailed reconstructions, but overfits to particular views and does not interpolate smoothly. SIMONe-VS on the other hand finely reproduces the object structure of the room. Even when it observes objects at a distance or up close, it places and sizes them correctly in totally novel views. This makes SIMONe-VS a powerful choice over NeRF-VAE and GQN-style models when the priority is to capture scene structure across diverse examples.

## 4.2 Instance segmentation (fully unsupervised)

Having shown the benefit of view information to inferring scene structure in the Section 4.1, we now turn to the added challenge of inferring viewpoint directly and simultaneously with scene contents (without any supervision).

|  | Static ARI-F | | | Video ARI-F | | | |
| --- | --- | --- | --- | --- | --- | --- | --- |
|  | MONet | SA | S-IODINE | MONet | SA | S-IODINE | SIMONe |
| Objects Room 9 | 0.886 | 0.784 | 0.695 | 0.865 | 0.066 | 0.673 | 0.936 |
|  | (±0.061) | (±0.138) | (±0.007) | (±0.007) | (±0.014) | (±0.0.002) | (±0.010) |
| CATER | 0.937 | 0.923 | 0.728 | 0.412 | 0.073 | 0.668 | 0.918 |
|  | (±0.004) | (±0.076) | (±0.032) | (±0.012) | (±0.006) | (±0.033) | (±0.036) |
| Playroom | 0.647 | 0.653 | 0.439 | 0.442 | 0.059 | 0.356 | 0.800 |
|  | (±0.012) | (±0.024) | (±0.009) | (±0.010) | (±0.002) | (±0.006) | (±0.043) |

Table 1: **SIMONe segmentation performance** (in terms of Adjusted Rand Index for foreground objects, ARI-F) compared to state-of-the-art unsupervised baselines: two static-frame models (MONet and Slot Attention, SA) and a video model (S-IODINE). We calculate static and video ARI-F scores separately. For static ARI-F, we evaluate the models per still image. For video ARI-F, we evaluate the models across space and time, taking an object's full trajectory as a single class. The video ARI-F thus penalizes models (especially Slot Attention) which fail to track objects stably. We report the mean and standard deviation of scores across 5 random seeds in each case.

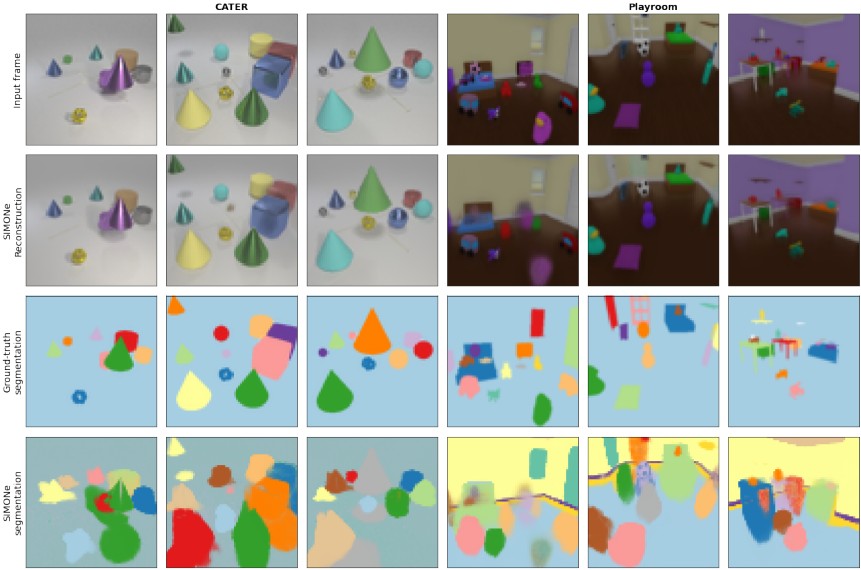

Figure 4: **Segmentations** and reconstructions produced by SIMONe on CATER and Playroom. SIMONe copes well with clutter and different-sized objects. It learns to use object motion as a segmentation signal on CATER, evident from the fact that an object's shadow is correctly assigned to that object's segment as it moves. This is true even when there's multiple shadows per object (due to multiple lights in the scene). SIMONe also overcomes color-based cues to segment two-toned objects such as beds in the Playroom as single objects. See Appendix A.5.2 to compare with baseline models.

We compare SIMONe to a range of competitive but viewpoint-unaware scene decomposition approaches. First, we train two static-frame models: MONet and Slot Attention. MONet uses a similar generative process and training loss to our model, achieving segmentation by modeling the scene as a spatial mixture of components, and achieving disentangled representations using a $\beta$-weighted KL information bottleneck. On the other hand, it uses a deterministic, recurrent attention network to infer object masks. Slot Attention is a transformer-based autoencoding model which focuses on segmentation performance rather than representation learning. Finally, we also compare against Sequential IODINE ("S-IODINE"), which applies a refinement network to amortize inference over time, separating objects by processing them in parallel. It also uses a $\beta$-weighted KL loss to disentangle object representations. Note that S-IODINE is a simplified version of OP3 [17], which additionally attempts to model (pairwise) object dynamics using an agent's actions as inputs. SIMONe and S-IODINE both avoid relying on this privileged information. Table 1 contains a quantitative comparison of segmentation performance across these models, while Figure 4 shows qualitative results.

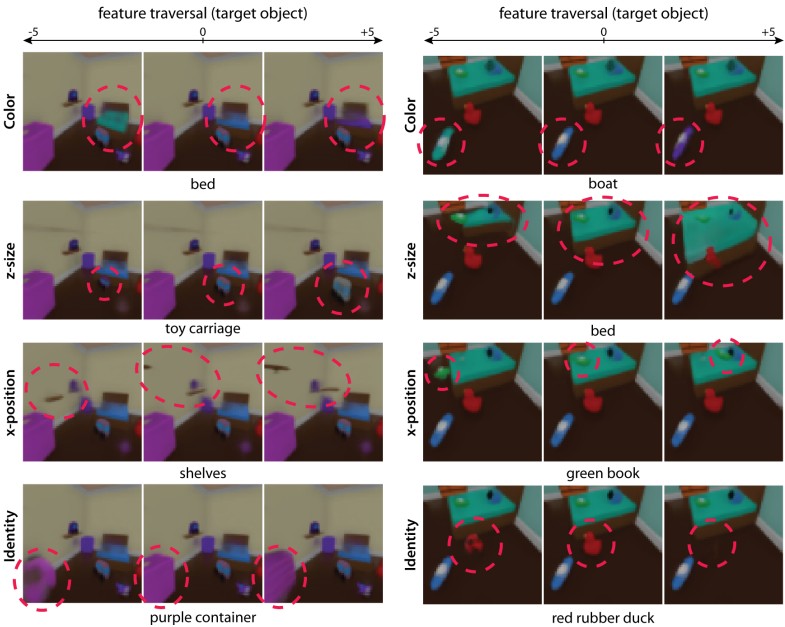

Figure 5: **Object attributes learnt by SIMONe.** In each row, we manipulate a particular object latent attribute for an arbitrary target object (circled in red) in two scenes. This reveals the attributes' relationship to interpretable object characteristics like color, size, position and identity.

# 5   Analysis

We take a closer look at the representations learnt by our model by decoding them in various ways. First, we manipulate latent attributes individually to assess the interpretability of object representations visually in Section 5.1. Next, we exploit SIMONe's latent factorization to render views of a given scene using the camera trajectory of a different input sequence. These cross-over visualizations help identify how the model encodes object dynamics in Section 5.2. Finally, we measure the predictability of ground-truth camera dynamics and object dynamics from the two types of latents in Section 5.3. These analyses use a single, fully unsupervised model per dataset.

## 5.1   Latent attribute traversals

We visualize the object representations learnt by SIMONe on Playroom to highlight their disentanglement, across latent attributes and across object slots, in Figure 5. We seed all latents using a given input sequence, then manipulate one object latent attribute at a time by adding fixed offsets.

Note that object position and size are well disentangled in each direction. Aided by the extraction of view-specific information in the frame latents, SIMONe also learns object features corresponding to identity. The decoder nevertheless obeys the biases in the dataset–for instance, shelves will slide along a wall when their position latent is traversed. The rubber duck does not morph into a chest of drawers because those are always located against a wall. This further suggests a well-structured latent representation, which the decoder can adapt to.

## 5.2   Object and frame latent cross-overs

We expect SIMONe to encode object trajectories and camera trajectories independently of each other. In fact, each object's trajectory should be summarized in its own time-invariant latent code. To examine this, we recompose object latents from one sequence with frame latents from other sequences in the CATER dataset. The result, in Figure 6, is that we can observe object motion trajectories from multiple camera trajectories.

Note the consistency of relative object positions (at any time-step) from all camera angles. In the single moving object case, its motion could in fact be interpreted as a time-varying global property of the scene. Despite this challenge, SIMONe is able to encode the object's motion as desired in its

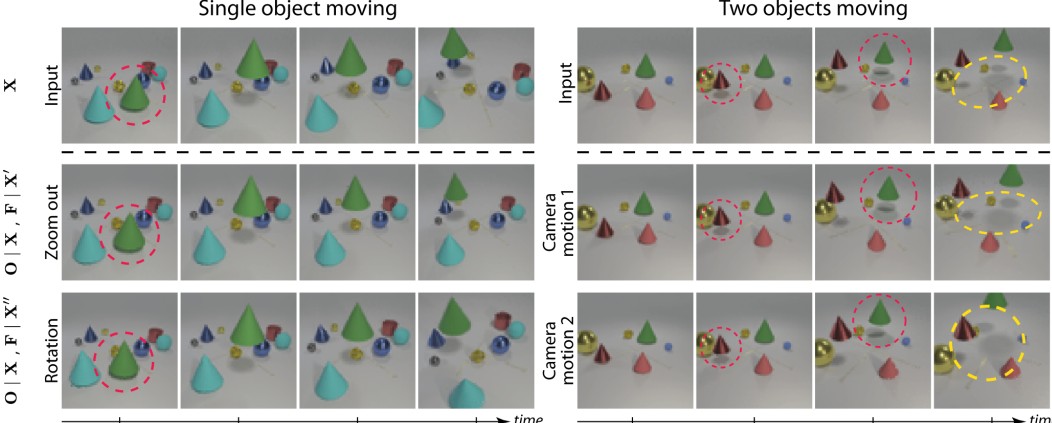

Figure 6: **Separation of object trajectories from camera trajectories. Left:** When encoding a sequence with consistent (i.i.d.) object dynamics, this information is extracted in the object latents and is unaffected by changing frame latents (see green cone). **Right:** Movement events are sequenced correctly; object relative positions also remain consistent (see pattern of shadows on the floor circled in yellow). See Appendix A.5.3 for cross-over plots showing more object trajectories.

specific time-invariant code. In Section 5.3, we further confirm that object trajectories are summarized in the object latents, which can be queried with time to recover allocentric object positions.

### 5.3 Camera pose and object trajectory prediction

We assessed SIMONe's frame latents by decoding the true camera position and orientation from them. We trained linear and MLP regressors to predict the camera pose at time $t$ from the corresponding frame latent $\mathbf{f}_t$ on a subset of CATER sequences. We also trained an MLP on the time-invariant object latents $\mathbf{o}_{1:K}$ for the same task. We evaluated these decoders on held-out data. Table 2 shows that frame latents describe the viewpoint almost perfectly.

We also assessed if the object latents contain precise information about allocentric object positions (to be clear, position information is not provided in any form while training SIMONe). Table 3 shows that the correct object latent is predictive of the allocentric position of a dynamic object (when queried along with the timestep). Adding the frame latent does not provide more information, and using

|  | Linear($\mathbf{f}_t$) | MLP($\mathbf{f}_t$) | MLP($\mathbf{o}_1, ..., \mathbf{o}_K$) |
|---|---|---|---|
| Camera location | $0.832 \pm 0.0$ | $0.949 \pm 0.002$ | $0.044 \pm 0.026$ |
| Camera orientation (Rodrigues) | $0.800 \pm 0.0$ | $0.946 \pm 0.002$ | $0.292 \pm 0.025$ |

Table 2: **Decoding camera pose.** We show that ground-truth camera location or orientation is predictable from the corresponding frame latent, but cannot be predicted from all object latents put together. We report the test $R^2$ score across 5 independently trained decoders per input type.

|  | MLP($\mathbf{o}_k$) | MLP($\mathbf{o}_k, t$) | MLP($\mathbf{o}_k, \mathbf{f}_t, t$) | MLP($\{\mathbf{o}_j : j \neq k\}$) |
|---|---|---|---|---|
| Trained on all objects | $0.710 \pm 0.006$ | $0.871 \pm 0.006$ | $0.876 \pm 0.003$ | $-0.062 \pm 0.006$ |
| Trained on moving objects | $0.724 \pm 0.007$ | $0.894 \pm 0.004$ | $0.898 \pm 0.005$ | $-0.022 \pm 0.025$ |

Table 3: **Decoding object trajectories.** We test MLP decoders on predicting allocentric object positions (of moving object in unseen scenes) based on the following inputs: (a) the corresponding object latent, (b) the timestep as well, and (c) the frame latent corresponding to that timestep as well, and (d) remaining object latents from the scene (not pertaining to the object of interest). The decoders were trained on arbitrary objects or a subset containing moving objects only. We report the test $R^2$ score across 5 independently trained decoders per input type.

the "wrong" objects (from the same scene) is completely uninformative. To perform this analysis, we needed to align SIMONe's inferred objects with the ground-truth set of objects. We used the Hungarian matching algorithm on the MSE of inferred object masks and ground-truth object masks to perform the alignment. Given SIMONe's disentangling of object dynamics, its time-abstracted object representations could prove helpful for a variety of downstream tasks (e.g. "catch the flying ball!").

Taken together, Table 2 and Table 3 show the separation of information that is achieved between the object and frame latents, helping assert our two central aims of view-invariant and temporally abstracted object representations.

# 6   Discussion and Future Work

**Scalability.** The transformer-based inference network in SIMONe makes it amenable to processing arbitrarily large videos, just as transformer-based language models can process long text. SIMONe could be trained on windows of consecutive frames sampled from larger videos (aka "chunks"). For inference over a full video, one could add memory slots which carry information over time from one window to the next. Applying SIMONe on sliding windows of frames also presents the opportunity to amortize inference at any given time-step if the windows are partially overlapping (so the model could observe every given frame as part of two or more sequences). Our use of the standard transformer architecture also makes SIMONe amenable to performance improvements via alternative implementations.

**Limitations. (1**) SIMONe cannot generate novel videos (e.g. sample a natural camera trajectory via consecutive frame latents) in its current version. This could be addressed in a similar fashion to the way GENESIS [5] built on MONet [3]–it should be possible (e.g. using recurrent networks) to learn conditional priors for objects in a scene and for successive frame latents, which would make SIMONe fully generative. **(2**) We see another possible limitation arising from our strict latent factorization. We have shown that temporally abstracted object features can predict object trajectories when queried by time. This can cover a lot of interesting cases (even multiple object-level "events" over time), but will start to break as object trajectories get more stochastic (i.e. objects transition considerably/chaotically through time). We leave it to future work to explore how temporal abstraction can be combined with explicit per-step dynamics modeling in those cases. For simpler settings, our approach to encoding object trajectories (distilling them across time) is surprisingly effective.

# 7   Conclusion

We've presented SIMONe, a latent variable model which separates the time-invariant, object-level properties of a scene video from the time-varying, global properties. Our choice of scalable modules such as transformers for inference, and a pixel-wise decoder, allow the model to extract this information effectively.

SIMONe can learn the common structure across a variety of procedurally instantiated scenes. This enables it to recognize and generalize to novel scene instances from a handful of correlated views, as we showcased via 360-degree view traversal in the view-supervised setting. More significantly, SIMONe can learn to infer the two sets of latent variables jointly without supervision. Aided by cross-frame spatio-temporal attention, it achieves state-of-the-art segmentation performance on complex 3D scenes. Our latent factorization (and information bottleneck pressures) further help with learning meaningful object representations. SIMONe can not only separate static object attributes (like size and position), but it can also separate the dynamics of different objects (as time-invariant localized properties) from global changes in view.

We have discussed how the model can be applied to much longer videos in the future. It also has potential for applications in robotics (e.g. sim-to-real transfer) and reinforcement learning, where view-invariant object information (and summarizing their dynamics) could dramatically improve how agents reason about objects.

## Acknowledgements

We thank Michael Bloesch, Markus Wulfmeier, Arunkumar Byravan, Claudio Fantacci, and Yusuf Aytar for valuable discussions on the purview of our work. We are also grateful for David Ding's support on the CATER dataset. The authors received no specific funding for this work.

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
