# A Appendix

A brief note on notation and convention—we've adopted the following standards for consistency across the paper:

1. Tensor axis order: when indexing into any output of our model (e.g. $\boldsymbol{\mu}_{k,t,i}$), the component aka object dimension (generally size K) precedes the time dimension (size T), which in turn precedes the spatial dimension. We use a single spatial dimension to account for 2D image-centric space.

2. Allocentric spatial axes: when discussing the position or orientation of an observer or object, we label the axes of 3D space consistently across datasets. The 'x' and 'z' directions span the ground plane (with z pointing determining an object's depth), while 'y' points upward (determining height).

3. Latent vs latents: We use "latent" to denote a multidimensional hidden variable. Therefore, "latents" denotes a *set* of multidimensional hidden variables (such as $\mathbf{O}$ or $\mathbf{F}$). Any particular dimension of a latent is referred to as a latent attribute or feature.

4. Segmentation maps: we use a consistent color scheme when plotting segmentation maps (e.g. Figure 4). See Figure 7 for our component-wise color palette. The component order is always determined by the output of a given model.

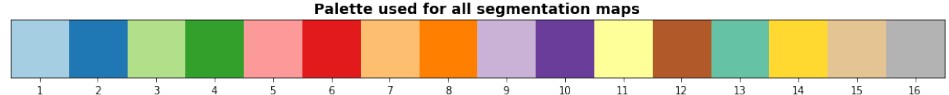

Figure 7: **Color palette** used for all segmentation masks.

## A.1 Extended Related Work

In Figure 8, we demonstrate a limitation of existing scene decomposition approaches which attempt to recognize object-based structure without taking into account a potentially moving observer.

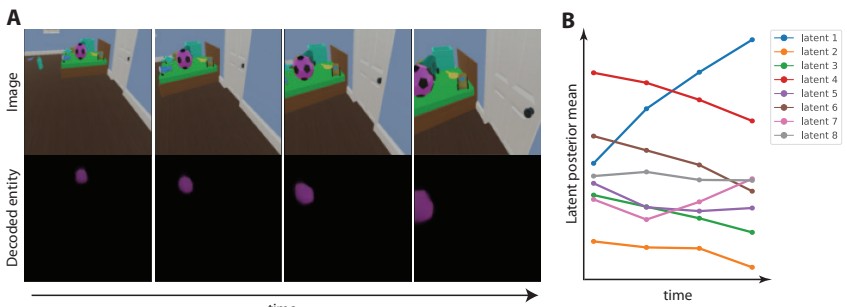

Figure 8: **Changing view in a 3D scene illustrating an issue with existing models. A.** Consider a static 3D scene observed from a changing viewpoint. We draw attention to the soccer ball lying on the bed. Its size and position appear to change in image space. Row 2 shows the decoded reconstruction of this particular object by a MONet [3] model (fed and trained with ground-truth object masks instead of inferring its own segmentation). **B.** Given their ignorance of spatial structure with respect to viewpoint, many existing object-centric representation learning methods are bound to conflate changes in viewpoint with object attributes; in this case, MONet encodes the changes directly into the ball's latent representation.

We also draw connections to more prior work (following Section 2 in the main paper):

**Video processing and vision.** (**1**) Human pose estimation has been a common motivation of view-invariant representation learning in vision [48–51]. Some of this work also uses variational unsupervised learning, but the focus is largely on single objects, or makes strong assumptions about

the kind of object (e.g. hands and their degrees of freedom [52]) being studied. (**2**) A subset of video understanding and generation models do aim to separate time-invariant and time-varying information like SIMONe. Some [53, 54] require specially designed objectives to achieve this separation; others [55–57] structure their generative models to encourage it. While this principle has enabled models to generate quality video predictions, all of them represent the whole scene with a single latent; they cannot decompose it into objects. (**3**) Another spate of successes has emerged from the use of transformers [58] in vision. They work well at supervised image and video tasks ranging from classification to detection and segmentation [59–62]. One such model [63] is capable of foreground-background segmentation without supervision but is trained on still images. For video-based tasks, [64–67] showed the importance of spatio-temporal attention (i.e., integrating information *jointly* over space and time), a principle that also works for SIMONe's inference network. However, most prior work relies on supervised learning. To our knowledge, SIMONe is the first model to demonstrate the benefit of spatio-temporal, cross-frame (rather than sequential) attention to decomposing multi-object scenes in a fully unsupervised manner.

## A.2 Datasets

### A.2.1 Objects Room 9

Objects Room was a MuJoCo-based dataset originally released [68, 44] under the Apache 2.0 license and used for prior work such as GQN and MONet [7, 3]. Our variant, which we denote Objects Room 9, contains more objects per scene (nine rather than three). We use length-16 input sequences with the camera moving on a fixed ring and facing the center of the room. The objects themselves are static.

### A.2.2 CATER

CATER (a dataset for Compositional Actions and TEmporal Reasoning) was released by [45] under the Apache 2.0 license. We augmented the open-source data generation scripts to further export ground-truth object masks (with lighting disabled, so there's no object shadows). We keep all settings identical to the publicly available version of the dataset containing the moving camera and two moving objects per scene. Like the original dataset, we have three, randomly placed light sources in each scene. This often leads to multiple shadows per object.

See Figure 9 for a scatter plot of ground-truth object positions in CATER, highlighting the presence of static and moving objects.

To train SIMONe, we crop the original 320x240 images centrally to a square aspect ratio and then resize them to 64x64. We use length-16 sequences from the beginning of each video. For segmentation figures on CATER (Figures 4, 16-18), we add a constant value 0.2 to images of the scene (and reconstructions) to increase their brightness. This is done for visualization only.

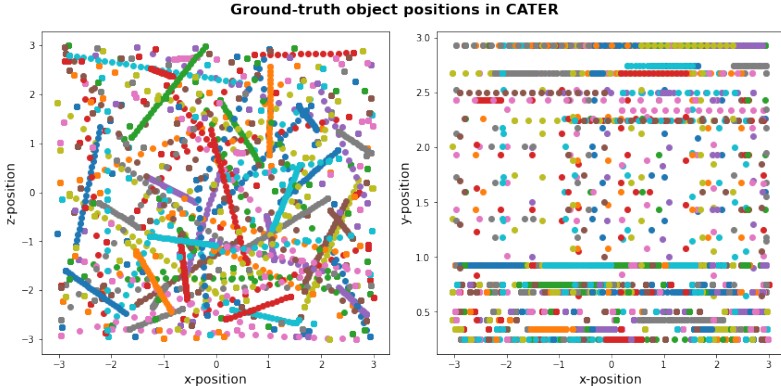

Figure 9: **Random sample of object positions** from 50 CATER scenes over 32 timesteps.

### A.2.3 Playroom

The Playroom is a Unity-based environment for object-centric tasks [69, 46], originally released as pre-packaged Docker containers with an Apache 2.0 license. We used an arbitrary behaviour policy (trained by demonstrations) to generate video sequences from the environment (one per episode). The L shape of the room remains the same in each instantiation but its appearance and all object/furniture attributes are varied. See Figure 10 for a sample of the agent's pose in the environment. See also Figure 11 for images of Playrooms arranged by the agent's orientation (yaw)—these can be used as a reference for the view synthesis outputs in Figures 3, 13, and 14.

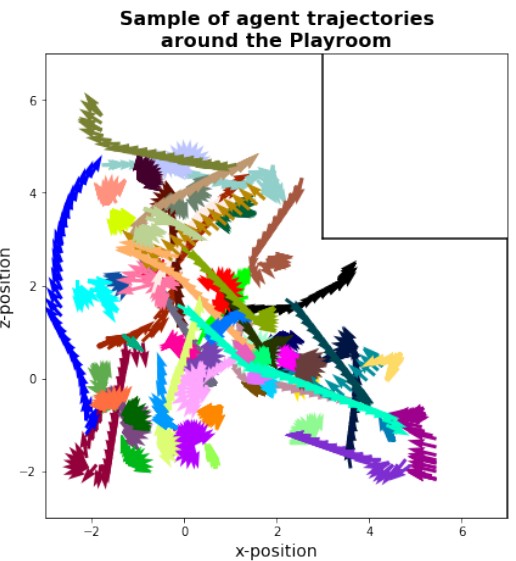

Figure 10: **Random sample of 64 ground-truth agent trajectories** around the Playroom. Arrow heads denote the agent's orientation, while arrow positions denote location. Note that the trajectories are varied, and the agent rarely observes the full room in a single sequence.

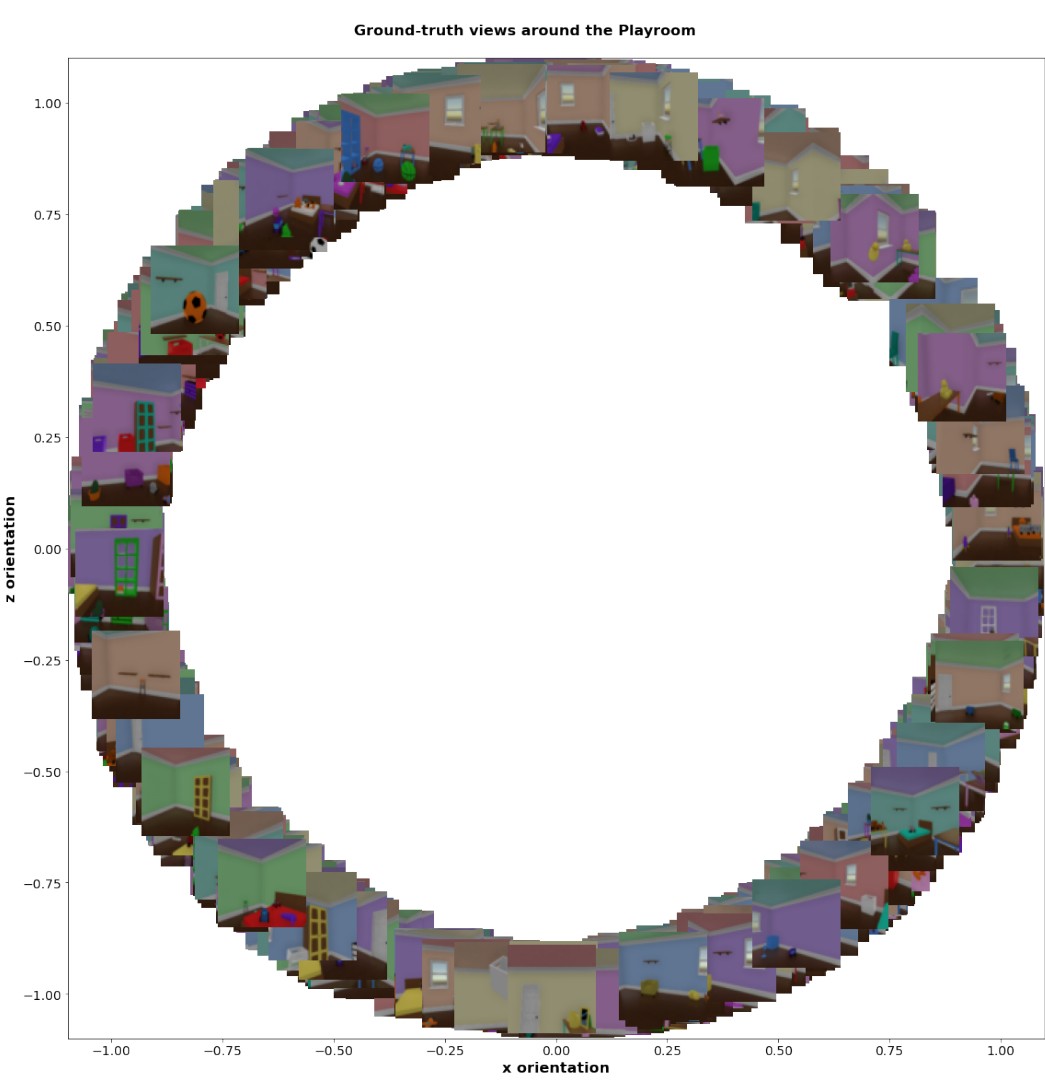

Figure 11: **A scatter plot of ground-truth views** around the Playroom, when the agent is roughly at the center of the room, with a level pitch (i.e. it's not looking at the floor or ceiling). This reveals the structure and variety in the procedurally generated Playroom from different orientations (yaw).

## A.3 Model and Hyperparameters

See Table 7 for compute resources used in training SIMONe and baseline models.

### A.3.1 Decoder

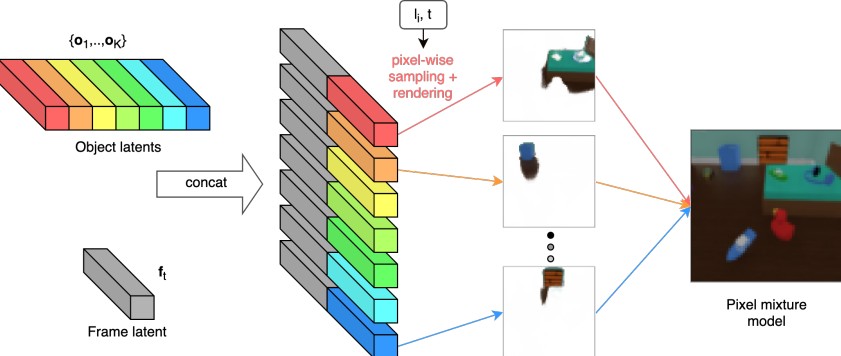

Figure 12: **SIMONe's decoder network** $\mathcal{D}_\theta$ architecture.

Our pixel-level decoder $\mathcal{D}_\theta(\mathbf{o}_k, \mathbf{f}_t; \mathbf{l}_i, t)$ is implemented as a 1x1 CNN (effectively a pixel-wise MLP), which receives concatenated inputs $[\mathbf{o}_k, \mathbf{f}_t, \mathbf{l}_i, t]$. Note this means the decoder can be parallelized (i.e. batch-applied) across all batch elements, all K objects, and all T frames. For CATER and Playroom, the CNN has 6 hidden layers with 512 output channels each. For Objects Room 9, we use 4 hidden layers with 128 channels each. An additional output layer produces 4 channels containing the RGB pixel (reconstruction) means $\boldsymbol{\mu}_{k,t,i}$ and mixture logits $\hat{m}_{k,t,i}$.

We've used $\mathcal{D}_\theta(\mathbf{o}_k, \mathbf{f}_t; \mathbf{l}_i, t)$ as shorthand for $\mathcal{D}_\theta(\mathbf{o}_{k,t,i}, \mathbf{f}_{k,t,i}; \mathbf{l}_i, t)$. Instead of prior decoding approaches [70] which sampled a single latent variable per object and broadcast it spatially to form the input to the decoder, we take i.i.d samples $\mathbf{o}_{k,t,i} \sim q(\mathbf{o}_k|\mathbf{X})$ across all time-steps $1 \leq t \leq T$ and across all pixels $1 \leq i \leq HW = 64 \cdot 64$. Likewise, we take i.i.d samples $\mathbf{f}_{k,t,i} \sim q(\mathbf{f}_t|\mathbf{X})$ across all object slots $1 \leq k \leq K$ and across all pixels $1 \leq i \leq HW = 64 \cdot 64$. This "multisampling" approach, while not crucial, does improve early training performance. We hypothesize that this is due to a reduction in the gradient bias resulting from (the alternative approach of) copying single samples $\mathbf{o}_k$ and $\mathbf{f}_t$ across all space and time in a sequence.

Hence, to be explicit, the full pixel-wise decoder can be written as follows:

$$\hat{m}_{k,t,i}, \boldsymbol{\mu}_{k,t,i} = \mathcal{D}_\theta(\mathbf{o}_{k,t,i}, \mathbf{f}_{k,t,i}; \mathbf{l}_i, t)$$

### A.3.2 Encoder

See Figure 2 in the main text for an architecture diagram.

**CNN.** The encoder $\mathcal{E}_\phi(\mathbf{X})$ contains an initial CNN which outputs $IJ = 8 \cdot 8$ spatial feature maps for each frame. The CNN layers have stride 2, kernel size 4, and output 128 channels each—hence the number of layers is determined by the size of the input (e.g. three layers when the input size $HW = 64 \cdot 64$). We use a ReLU activation after each layer.

**Transformers.** The CNN is followed by transformers $\mathcal{T}_1$ and $\mathcal{T}_2$, which have identical hyperparameters. For CATER and Playroom, the transformers have 4 layers, 5 heads, value size 64, and MLPs with a single hidden layer (size 1024). For Objects Room 9, we use even simpler transformers, with 3 heads and 256 MLP hidden units each (other settings are kept the same). Note that the transformer embedding size is determined as the product of the number of heads and the value size—it is not constrained by the dimensionality of the transformer inputs.

**Spatial pool (if necessary).** In between $\mathcal{T}_1$ and $\mathcal{T}_2$, we need to reduce the number of slots from $TIJ$ to $TK$ (if $IJ > K$). We use a strided *sum* across the spatial dimensions (sized $I$ and $J$) to do so, with the kernel size and stride each set to $[I/\sqrt{K}, J/\sqrt{K}] = [2, 2]$. Finally, we scale the pooled values by $\sqrt{K/IJ} = \sqrt{16/64} = 1/2$ for our experiments.

|  | **Objects Room 9** | **CATER** | **Playroom** |
|---|---|---|---|
| Shape of input, [T, H, W] | [16, 64, 64] | [16, 64, 64] | [32, 64, 64] |
| Shape of decoded image, $[T_d, H_d, W_d]$ | [4, 32, 32] | [8, 32, 32] | [4, 64, 64] |
| Number of objects, K | 16 | | |
| Frame/object latent dimensionality | 32 | | |
| Object latents KL loss weight, $\beta_o$ | 1e-8 | 1e-5 | Annealed: 5e-6 $\rightarrow$ 5e-7 (exponential window: 50k steps) |
| Frame latents KL loss weight, $\beta_f$ | 1e-8 | 1e-4 | 1e-7 |
| Reconstruction (NLL) loss scale, $\alpha$ | 0.2 | | |
| Pixel likelihood scale, $\sigma_x$ | 0.08 | | |
| Number of training iterations | 4e5 | 2.5e6 | 2.5e6 |
| Learning rate | 20e-5 | | |
| Optimizer | Adam | | |

Table 4: **Summary of hyperparameters** for SIMONe across all datasets.

**MLPs.** The outputs $\hat{e}_{k,t}$ of $\mathcal{T}_2$ are aggregated separately along the spatial and temporal axes. These aggregates are passed through $\text{mlp}_f$ and $\text{mlp}_o$ to yield the frame and object posterior parameters respectively. Both MLPs have a single hidden layer with 1024 units.

### A.3.3 Training details

While training, we decode fewer frames than the number fed and encoded by the model as motivated in Section 3.3. We simply take $T_d$ random frame indices from $\{1, ..., T\}$ without replacement. However, when we subsample pixels to decode, we use a strided slice of the input (e.g. stride [2, 2] when $H_d = H/2$ and $W_d = W/2$). Note that for any evaluation or visualization, we decode the full length and size $[T, H, W]$ of the input sequence.

### A.4 Baseline Models

See Table 7 for compute resources used in training baseline models.

### A.4.1 Slot Attention

For all three datasets, we use slot size (i.e., dimensionality) 32 and train for 500,000 steps with a batch size of 128. We use linear learning rate warmup over the first 10,000 steps. Slot Attention seemed very sensitive to the choice of number of slots, with the model prone to oversegmenting rather than leaving slots empty. So we swept over the following settings for all datasets: number of slots (7 vs 10), encoder/decoder architecture (see Table 5), and learning rate ($4e-3, 4e-4, 4e-5$). We ran 5 random seeds for each hyperparameter setting. The mean and standard deviations reported in the Table 1 are calculated over the random seeds for the best performing hyperparameter setting (in terms of ARI-F).

### A.4.2 MONet

For all three datasets, we use 10 object slots, 64-dimensional latents, and train for 5,000,000 steps with an effective batch size of 32. We swept over the following: scale of the KL penalty $\beta$ (0.5 or annealed from 0.01 to 30 with an exponential window of 200,000 steps), pixel likelihood scale for the foreground slots (0.08 or 0.09), and the size of the pixel broadcast decoder (64 channels/5 hidden layers or 512 channels/6 hidden layers). The decoder settings are the same we used for the Slot Attention sweeps, except that we use 1x1 convolutions with stride 1 for MONet. Note

| | | **Small** | **Large** |
|---|---|---|---|
| **Encoder** | CNN | Conv2D(c=64, k=5, s=1)
Conv2D(c=64, k=5, s=1)
Conv2D(c=64, k=5, s=1)
Conv2D(c=64, k=5, s=1) | Conv2D(c=512, k=5, s=1)
Conv2D(c=512, k=5, s=1)
Conv2D(c=512, k=5, s=1)
Conv2D(c=512, k=5, s=1)
Conv2D(c=512, k=5, s=1)
Conv2D(c=512, k=5, s=1) |
| | Position MLP | MLP(64) | MLP(512) |
| | Output MLP | MLP([64, 64]) | MLP([512, 512]) |
| | Slot Attention | num iterations=3, slot size=32
GRU(32)
MLP(128) | num iterations=3, slot size=32
GRU(32)
MLP(512) |
| **Decoder** | CNN | Conv2D$^T$(c=64, k=5, s=2)
Conv2D$^T$(c=64, k=5, s=2)
Conv2D$^T$(c=64, k=5, s=2)
Conv2D$^T$(c=64, k=5, s=1)
Conv2D$^T$(c=64, k=5, s=1)
Conv2D(c=4, k=3, s=1) | Conv2D$^T$(c=512, k=1, s=1)
Conv2D$^T$(c=512, k=1, s=1)
Conv2D$^T$(c=512, k=1, s=1)
Conv2D$^T$(c=512, k=1, s=1)
Conv2D$^T$(c=512, k=1, s=1)
Conv2D$^T$(c=512, k=1, s=1)
Conv2D(c=4, k=3, s=1) |
| | Position MLP | MLP([64, 32]) | MLP([128, 32]) |

Table 5: **Slot Attention baseline architectures.** c: number of channels, k: kernel size, s: stride, Position MLP: MLP applied to positional encoding. MLP([m, n]) denotes an MLP with layers of size m and n.

that the background pixel likelihood scale is fixed at 0.07. The encoder is identical to the original implementation.

### A.4.3 S-IODINE

For all three datasets, we use latent size of 64 and train for 500,000 steps with a batch size of 128. We use a fixed learning rate of 3e-4 using the Adam optimizer. We use 7 slots for all models. We swept over the following: encoder/decoder architecture (see Table 6), scale of KL term (0.5 or 1.0) and output likelihood standard deviation (0.08 and 0.09).

### A.4.4 GQN

For the view synthesis comparison, we trained GQN on Playroom data. The architecture is the same as in [30]. We use an autoregressive decoder with 5 steps, 16-dimensional latents, and 256 hidden units in each layer. We use a Nouveau ResNet encoder and a 2D convolutional LSTM as the recurrent core. We set the likelihood scale to 0.08 to ensure the output distribution is parameterized identically to SIMONe-VS. To train the model, we used GECO to target a minimum reconstruction log likelihood (4.3, 4.5 or 4.7 nats per pixel). We used 16 frames as the context for each scene; these were randomly sampled from a 32-frame sequence.

### A.4.5 NeRF-VAE

The NeRF-VAE model for the view synthesis comparison is the same as in [30]. Like GQN, we trained NeRF-VAE with GECO, setting thresholds of 3.8, 4.0 or 4.2 nats per pixel for the reconstruction log likelihood (values higher than 3.8 were not attained). We used 16 context frames from the sequence, 0.08 likelihood scale, and 512 latent dimensions.

## A.5 Extended Results

### A.5.1 View synthesis

Figures 13-14 show more examples of viewpoint traversals comparing SIMONe-VS, NeRF-VAE and GQN. We execute the same traversal around the room (as in Figure 3) given different input sequences

|  |  | **Small** | **Large** |
|---|---|---|---|
| Encoder | CNN | Conv2D(c=32, k=3, s=2)
Conv2D(c=32, k=3, s=2)
Conv2D(c=32, k=3, s=2)
Conv2D(c=64, k=3, s=2) | Conv2D(c=64, k=5, s=1)
Conv2D(c=64, k=5, s=1)
Conv2D(c=64, k=5, s=1)
Conv2D(c=64, k=5, s=1)
Conv2D(c=64, k=5, s=1) |
|  | Output MLP | MLP([256, 256]) | |
|  | Refinement Network | LSTM(128)
Linear(128) | |
| Decoder | CNN | $\text{Conv2D}^T$(c=32, k=5, s=2)
$\text{Conv2D}^T$(c=32, k=5, s=2)
$\text{Conv2D}^T$(c=32, k=5, s=2)
$\text{Conv2D}^T$(c=32, k=5, s=1)
Conv2D(c=4, k=3, s=1) | $\text{Conv2D}^T$(c=512, k=1, s=1)
$\text{Conv2D}^T$(c=512, k=1, s=1)
$\text{Conv2D}^T$(c=512, k=1, s=1)
$\text{Conv2D}^T$(c=512, k=1, s=1)
$\text{Conv2D}^T$(c=512, k=1, s=1)
Conv2D(c=4, k=3, s=1) |

Table 6: **S-IODINE baseline architectures.** c: number of channels, k: kernel size, s: stride, MLP([m, n]) denotes an MLP with layers of size m and n.

|  | **Playroom** | **CATER** | **Objects Room** |
|---|---|---|---|
| SIMONe | 64/128 TPUv1 × 9 sweeps × 5 reps | 64 TPUv2 × 4 sweeps × 5 reps | 64 TPUv2 × 4 sweeps × 5 reps |
| MONet | 32 TPUv1 × 8 sweeps × 5 reps | 32 TPUv1 × 8 sweeps × 5 reps | 32 TPUv1 × 8 sweeps × 5 reps |
| S-IODINE | 32 TPUv1 × 16 sweeps × 5 reps | 32 TPUv1 × 16 sweeps × 5 reps | 32 TPUv1 × 16 sweeps × 5 reps |
| Slot Attention | 8 TPUv2 × 24 sweeps × 5 reps | 8 TPUv2 × 24 sweeps × 5 reps | 8 TPUv2 × 24 sweeps × 5 reps |

Table 7: **Compute resources** used to train SIMONe and baseline models. TPU: Tensor Processing Unit. TPUv1 and TPUv2 have 8GiB and 16GiB memory respectively. Each TPU unit refers to 1 core. Each "sweep" refers to a unique hyperparameter combination, while "reps" refer to independent random seeds.

(i.e. different rooms observed partially via different agent trajectories). Note that some context input sequences contain very little motion between frames (see last example in Figure 14). The model still handles this well.

We also compare the view-supervised models based on their representation cost and reconstruction fidelity (rate and distortion respectively) in Figure 15. We first trained SIMONe-VS, and attempted to replicate its attained log-likelihood in the baseline models via constrained optimization (GECO). NeRF-VAE and GQN reconstructions saturate at the low end of the GECO thresholds we tried, implying bottlenecks that prevent the models from reaching higher log likelihoods. GQN does match SIMONe-VS in its KL representation cost (summed over K object latents in SIMONe-VS's case). But as observed in the view synthesis images, GQN overfits to the context frames and is unable to interpolate between viewpoints successfully.

This speaks to the effectiveness of SIMONe's object-centric latent structure. Generally speaking, we expect SIMONe to achieve better reconstruction fidelity at the same total KL cost (or a similar reconstruction fidelity at lower KL cost) than a model which infers a single latent variable for the whole scene.

### A.5.2   Instance segmentation

Figures 16-18 visualize how baseline models perform at segmenting CATER and Playroom scenes in comparison to the SIMONe results we showed in Figure 4. Note that for all segmentation maps visualized, we use soft (inferred) component mixture weights in combination with the color palette in Figure 7. This helps show each model's confidence in its segmentation. A blurry segmentation map (e.g. S-IODINE on Playroom data in Figure 17) suggests higher-entropy component weights.

To elaborate on the qualitative differences between the models, note that (**1**) MONet exhibits a tendency for clustering by color. On Playroom data, it consistently merges the brown base of each bed with the brown floor. It also groups toys/small objects by color (e.g. the blue mattress and blue-windowed bus in column 4 or the purple duck and purple cushion in column 5 of Figure 16). On

the other hand, SIMONe segments the full (two-colored) bed as one object consistently, and we don't see it group the aforementioned toys by color. **(2)** When Slot Attention segments CATER, it ignores object shadows completely. It infers crisp shapes, showing a clear propensity to use each object's uniform color and lack of texture. SIMONe on the other hand assigns every object's shadows (up to three) in the corresponding object's segment.

### A.5.3   Temporal abstraction and dynamics prediction

We show more examples of the separation between object representations/trajectories and the camera's trajectory on CATER in Figure 19. In particular, we showcase other object dynamics present in the dataset (e.g. objects sliding on the floor or rotating), which SIMONe also captures accurately. As before, the first row in each figure shows an input sequence $\mathbf{X}$. The other two rows reuse the object latents $\mathbf{O}|\mathbf{X}$ from the first sequence, but recompose them with frame latents from other (arbitrary) sequences: $\mathbf{F}|\mathbf{X}'$ and $\mathbf{F}|\mathbf{X}''$. We observe that the recomposed scenes are still composed of the same objects with their exact trajectories, while only the camera motion changes. This shows that object trajectories are represented invariantly of viewpoint (and vice versa).

### A.5.4   Frame latents

To expand on our quantitative assessment of frame latents in Table 2, we look at the effect of hyperparameters $\beta_o$ and $\beta_f$ on the relationship between frame latents and camera pose (see Figure 20). This relationship is one determinant of how "view-invariant" the object latents can possibly be.

Beyond their aggregate information content, we might also want that frame latents capture meaningful changes (e.g. in terms of view) per latent dimension. Figure 21 shows the effect of individual frame latent attributes on Playroom. We selected the top ranking latents by marginal KL, and traverse them individually on several seed scenes. We indicate our interpretations of their behaviour in each row. Note that the latents controlling position in the room appear somewhat entangled. This may be a consequence of the policy used to collect our dataset; the agent's position is not arbitrary but influenced by the objects in the room.

### A.5.5   Composition of object latents from different scenes

Given an object-centric representation, one should be able to manipulate scene contents and produce plausible compositional behaviour. This could involve removing or adding objects, swapping content between scenes, or varying the number of objects. We present an early assessment of SIMONe's capabilities to perform such scene editing. We take a few different input scenes and compose (subsets of) their object latents into a novel composite scene, which can be then rendered from different points of view. See Figure 22 for an example.

### A.6   Wider Impact

SIMONe could benefit robotics and computer vision in multi-object scenes, whether indoors or on the street. There is some potential for misuse, especially in surveillance. While some prior work [20] has in fact used CCTV visuals of crowded scenes to demonstrate real use cases, we refrained from doing so.

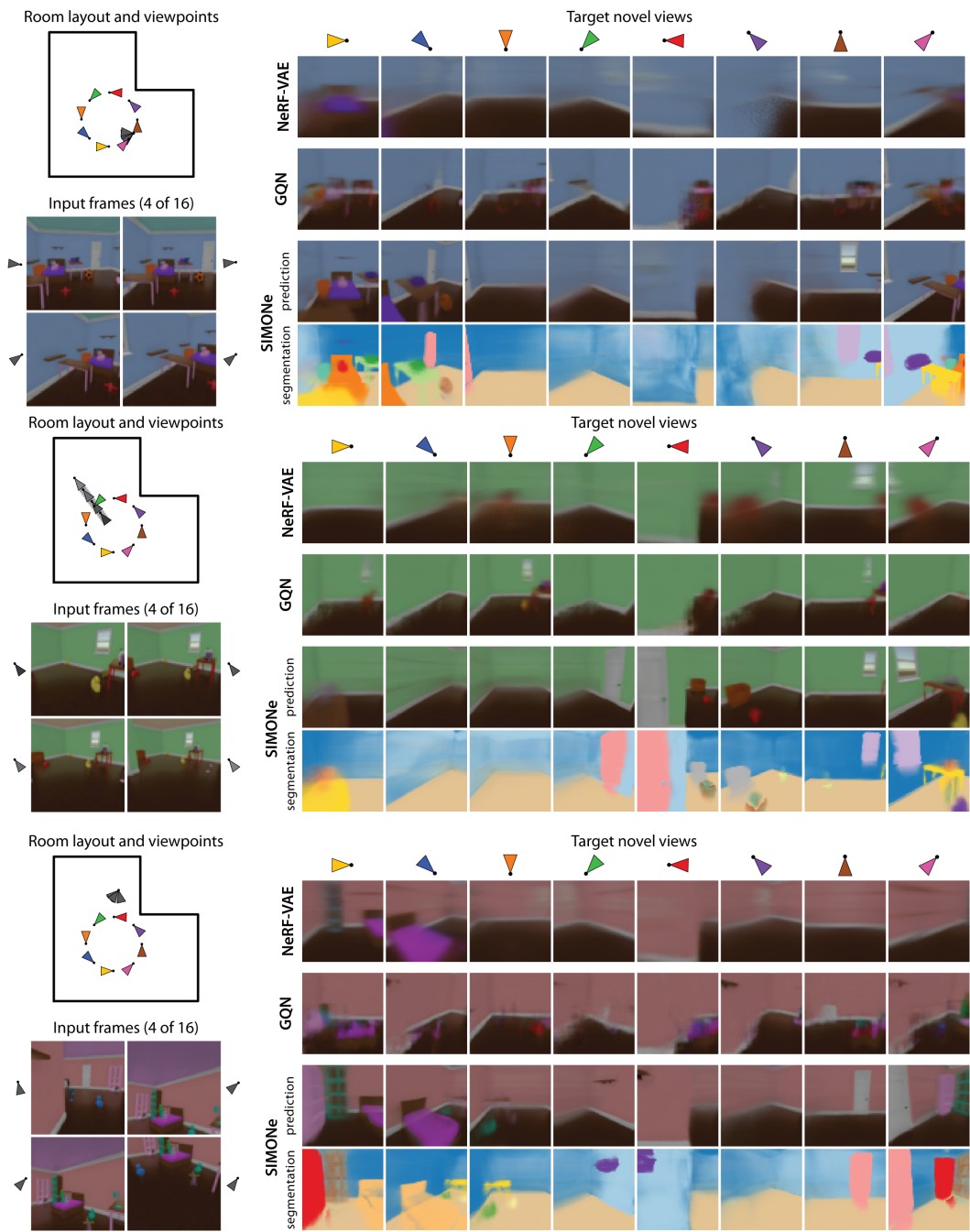

Figure 13: **Extended comparisons of scene representation and view synthesis capabilities** between SIMONe-VS, NeRF-VAE, and GQN. In each figure, a sub-sample of the input context sequence is shown at the lower left. At the upper left, a map of the room shows the camera pose corresponding to the visualized input images in dark gray, as well as the remaining input frames (observed by the models) in light gray. Columns on the right correspond to the colored (novel) viewpoints on the map. Refer to Figure 3 for more details.

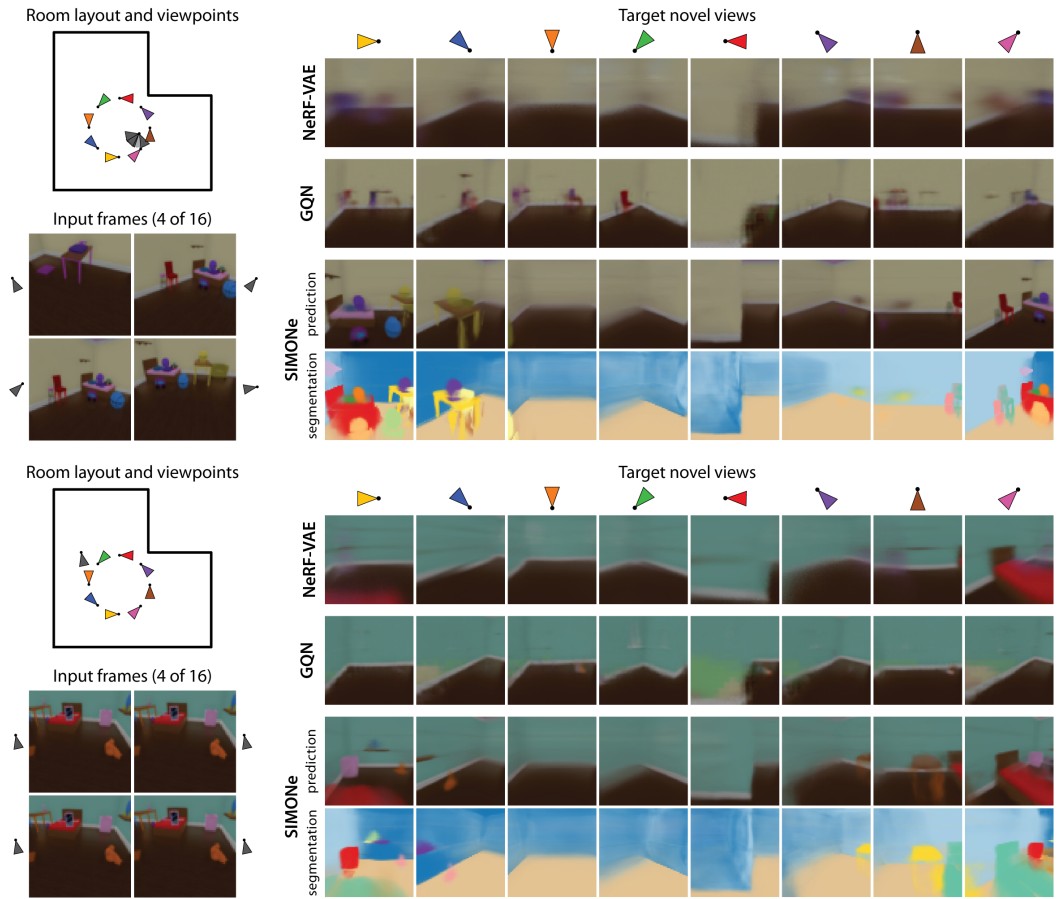

Figure 14: **Extended comparisons of scene representation and view synthesis capabilities** between SIMONe-VS, NeRF-VAE, and GQN. Refer to Figures 3, 13 for more details.

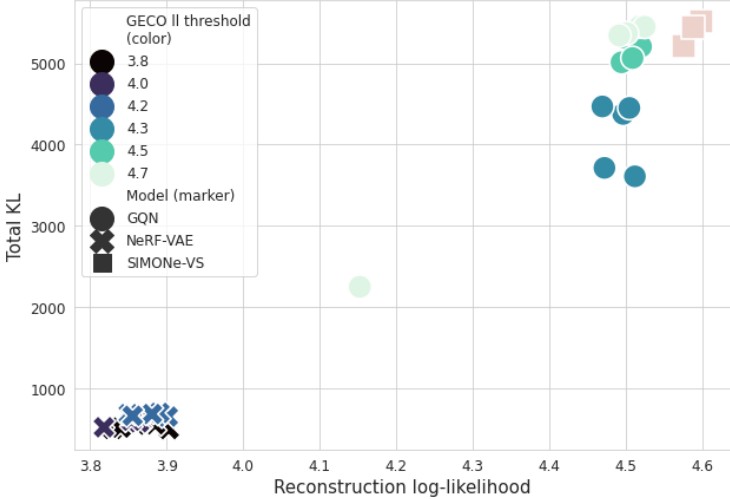

Figure 15: **Log-likelihood & KL comparison across view-supervised models.** We show five independent runs of GQN and NeRF-VAE for three GECO log-likelihood thresholds each. For SIMONe-VS, we show three independent runs (trained as usual without GECO). Note that SIMONe-VS's KL shown here is the sum (not average) over K object latents.

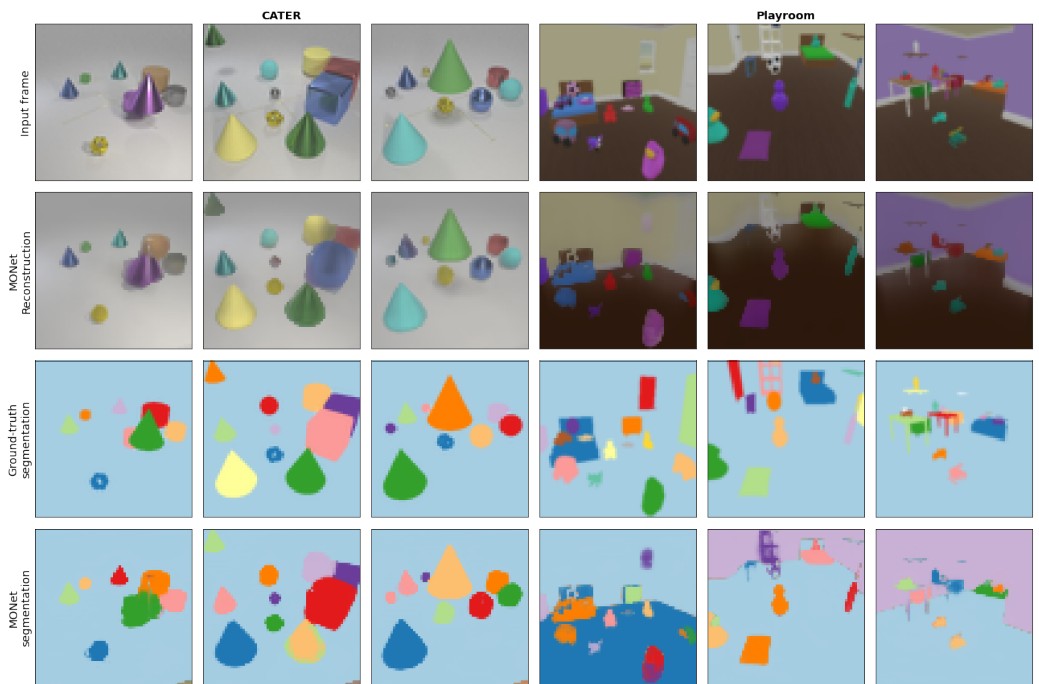

Figure 16: **Segmentations produced by MONet** on CATER and Playroom. Compare to Figure 4 in main text.

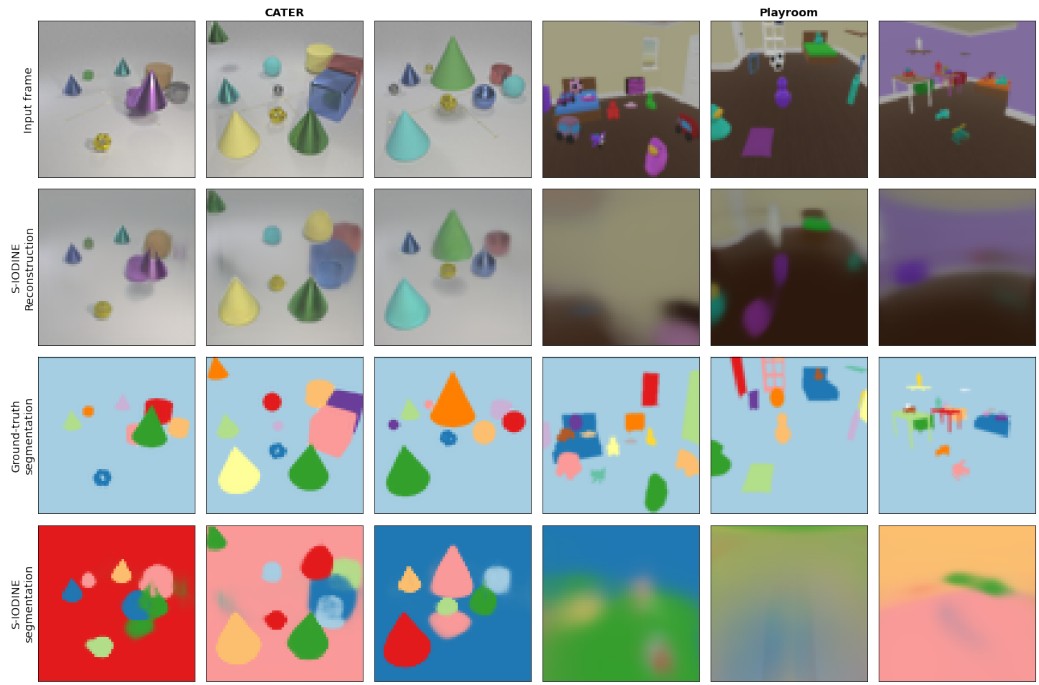

Figure 17: **Segmentations produced by S-IODINE** on CATER and Playroom. Compare to Figure 4 in main text.

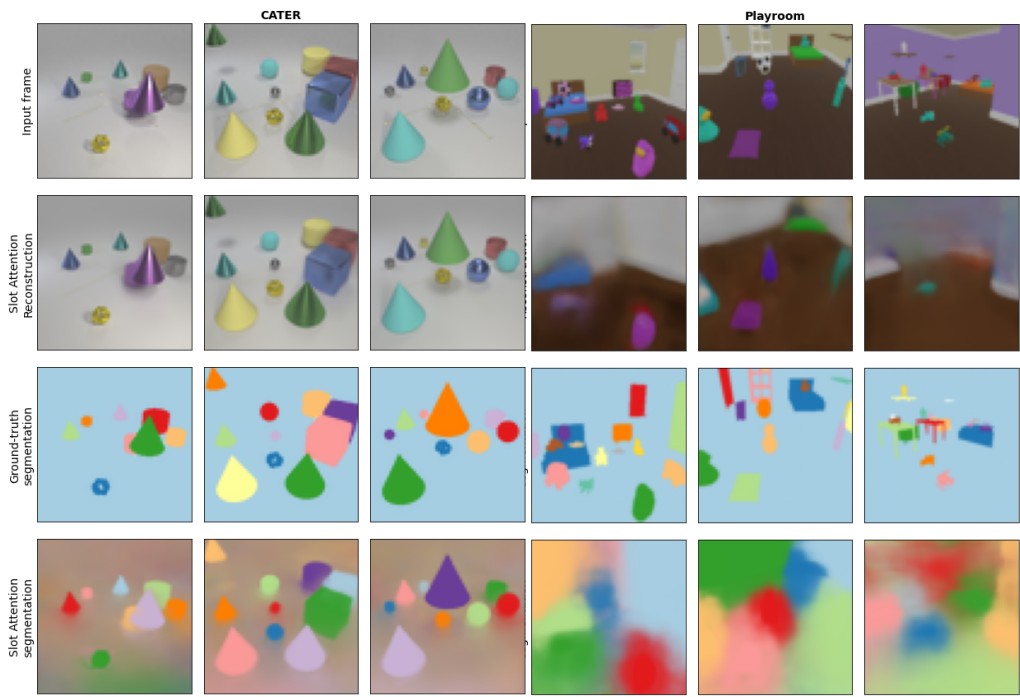

Figure 18: **Segmentations produced by Slot Attention** on CATER and Playroom. Compare to Figure 4 in main text.

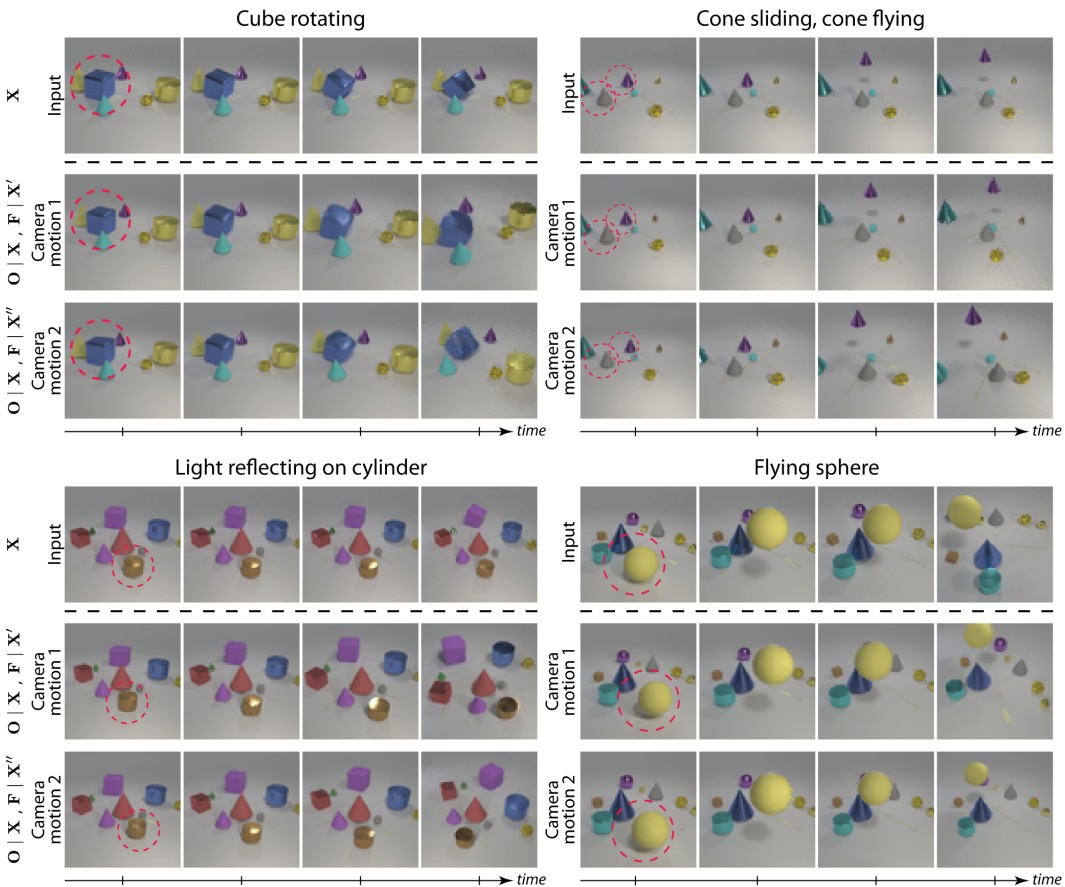

Figure 19: **Extended results showing separation of object trajectories from camera trajectories.** See Figure 6 for details.

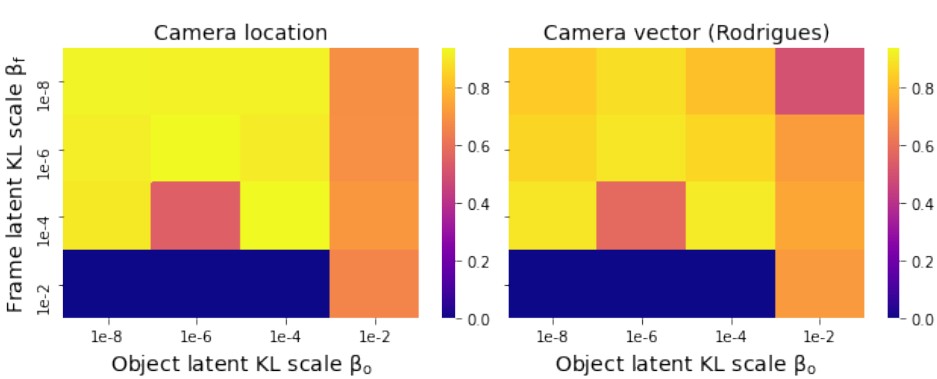

Figure 20: **Camera pose prediction given KL pressure hyperparameters.** Effect of $\beta_o$ and $\beta_f$ on the performance of predicting camera pose from frame latents ($R^2$ score, higher is better).

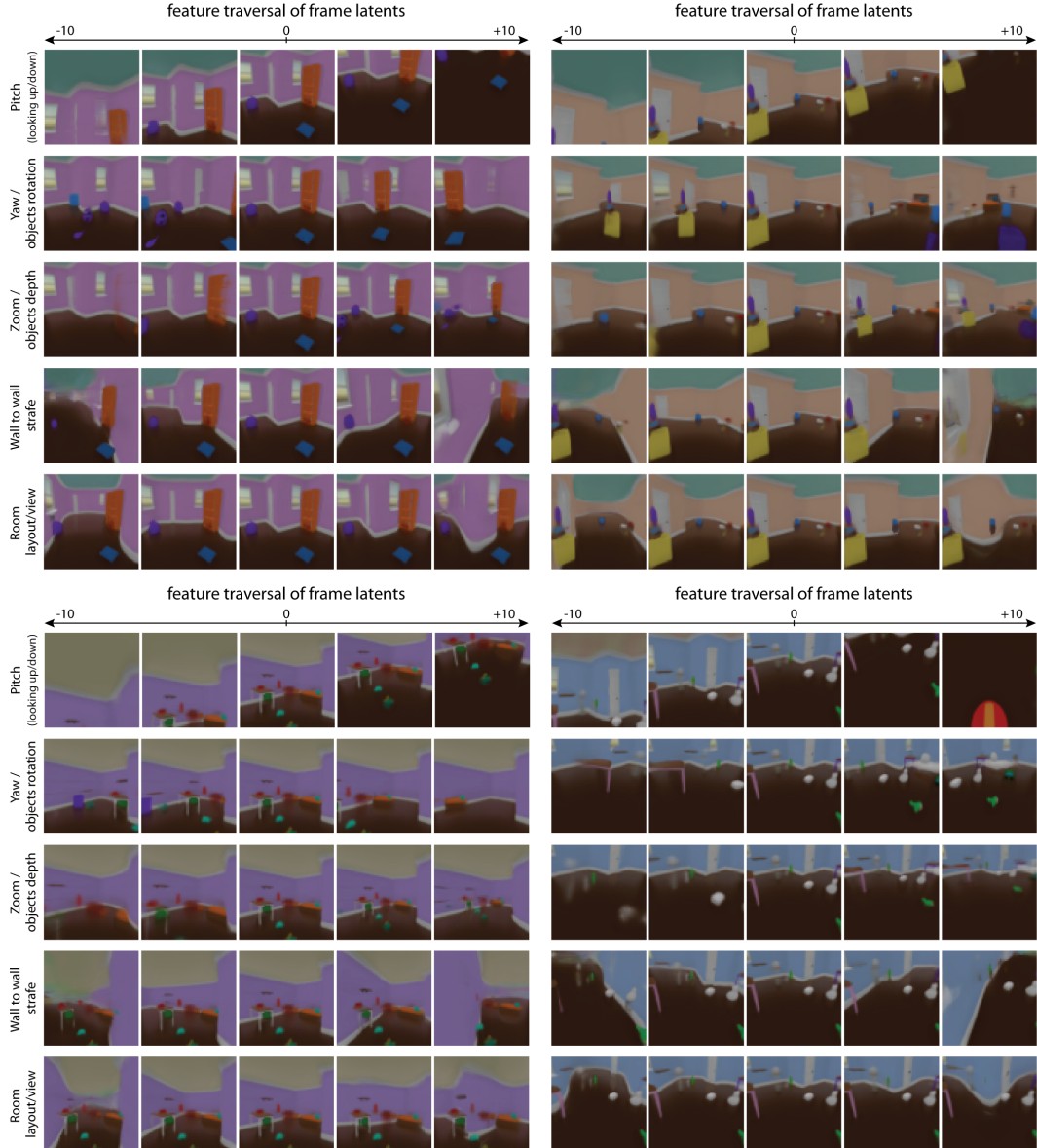

Figure 21: **Traversals of frame latent attributes.** We manipulate independent dimensions of a given frame latent. These should hypothetically control global/view attributes. (In contrast to Figure 5, all objects in the scene are affected when a frame latent attribute is manipulated). Each panel shows a different seed scene (visible in the middle column). The rows correspond to different latent attributes being manipulated (values are across the columns). We have labeled the rows with our interpretation for the effect of each latent attribute.

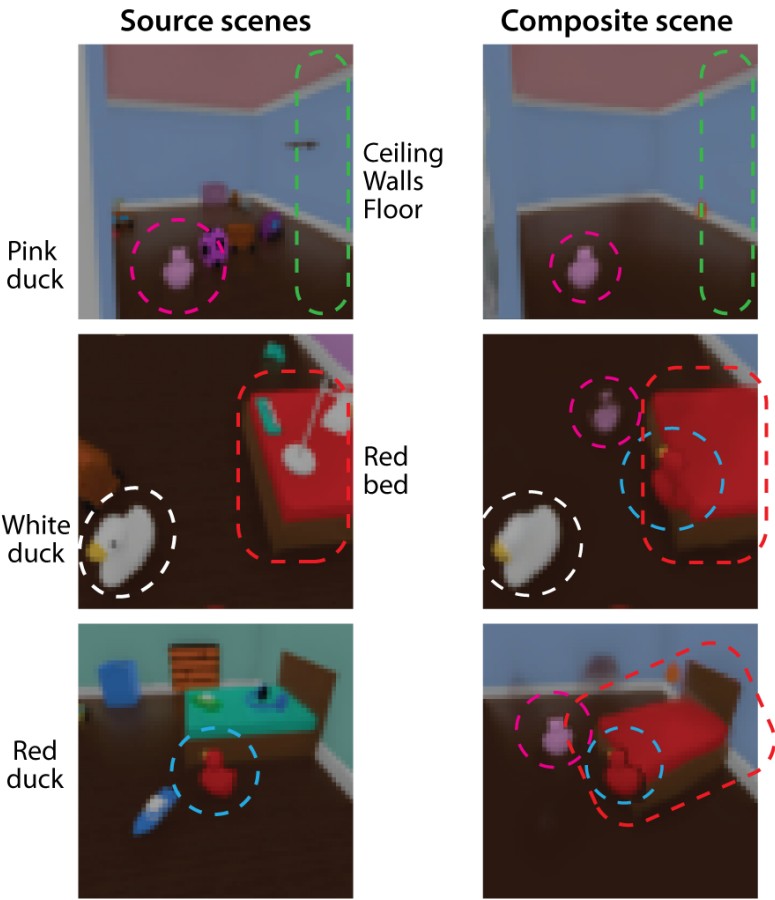

Figure 22: **Composition of object latents from different scenes. Left:** We take three different source scenes, and select different content from each (i.e. specific object latents, corresponding to the circled/labelled objects). **Right:** We then compose the selected contents into a novel set of objects (fewer than the usual $K = 16$), which we can render from any viewpoint around the room (for this figure, we take the frame latent $\mathbf{f}_t$ from each source image on the left). Note that the model is able to cope with removing and adding objects, and renders them in a plausible fashion despite never being trained to do so.