# OpenReview forum: "SIMONe: View-Invariant, Temporally-Abstracted Object Representations via Unsupervised Video Decomposition"
_NeurIPS.cc/2021/Conference — NeurIPS 2021 Poster_

### Official Review · Reviewer_5tHk · 2021-07-16

**Rating:** 7
**Confidence:** 3

**Summary:**

This paper proposes SIMONe, a variational auto-encoder that decomposes a video into object latents and frame latents without any supervision. The object latents correspond to the time-invariant, object-level contents of the scene. The frame latents correspond to the global time-varying elemnts of the scene such as viewpoint. Each pixel is decoded independently by querying a pixel-wise decoder using sampled latents. To infer latents, SIMONe uses a transformer-based encoder that integrates information jointly across space and time. The experiments demonstrate SIMONe's capabilities in view syntheses and instance segmentation. Furthermore, the object latents are shown to be disentangled and summarize the trajectories of the objects while the frame latents are shown to contain viewpoint information.

**Limitations And Societal Impact:**

Yes

**Main Review:**

Originality: This paper proposes a novel architecture that makes relatively few modeling assumptions about the scene when compared with previous works. The independence assumption in the decoder is also an interesting choice to allow the model to train on smaller inputs than the full image. It may be an interesting ablation to see the training speed tradeoff with this choice.

Quality: The model is sound and makes reasonable assumptions. The claims are well supported in the experiments. In particular, Table 2 demonstrates the camera location information is encoded in the frame latents Table 7 in the supplementary demonstrates the object trajectories are encoded in the object latents. The view synthesis and segmentation results also show that SIMONe outperforms the baselines. Since the authors ran their model on the CATER dataset, it would be interesting to see if the factorized representations can be used in the CATER localization task.

Clarity: The paper is clearly written and well organized. However, I did find that I had to look in the Supplementary to get a full understanding of the model, especially for the decoder. The authors may consider moving parts of the appendix to the main text if space allows. One minor suggestion is to include the spatial pooling between the two transformers in Figure 2, since that is an important operation in obtaining the K object latents. As drawn, it is not clear that K and IJ may be different.

Significance: Given the current state of unsupervised visual scene understanding models, this paper provides a good direction for future research. I would be curious how SIMONe performs on more realistic scenes than the ones shown in the paper.

**Time Spent Reviewing:**

12

---

> ### Author Response · Authors · 2021-08-06
> **Acknowledgement**
>
> Thanks for your positive review and an excellent summary of our work! We're glad you appreciated the different aspects of the paper, including the separation of information between the object and frame latents. We will respond to your questions and suggestions in detail very shortly. Please let us know if you have any further follow-ups for us (barring additional experiments as we’re not allowed to update the paper draft).

---

> ### Author Response · Authors · 2021-08-11
> **Main response**
>
> Thanks once again for spending more than a full day studying our work! We address the points you raised below.
>
> ### *The independence assumption in the decoder is also an interesting choice to allow the model to train on smaller inputs than the full image. It may be an interesting ablation to see the training speed tradeoff with this choice.*
> - The reviewer raises an excellent point. By decoding 1/4 of the full image, we were able to use a larger decoder (with more layers/parameters). We posit that could be a useful trick in applying the model to larger images or longer sequences (because it reduces the memory footprint substantially). We didn’t notice any significant change in reconstruction/segmentation performance on our datasets, but a full assessment of how much of the image can be discarded is left for further work.
> - Because we traded off pixels decoded for more parameters in the decoder, we didn’t benefit in terms of faster computation/training speed. But that is indeed a different possible trade-off.
>
> ### *Since the authors ran their model on the CATER dataset, it would be interesting to see if the factorized representations can be used in the CATER localization task*
> Indeed. Table 7 suggests SIMONe can localize CATER objects (moving ones in particular) to a highly accurate degree. While we haven’t trained on full-length CATER videos yet, it would be very interesting to confirm that our representations are immediately useful on the localization task.
>
> ### *The paper is clearly written and well organized. However I had to look in the Supplementary to get a full understanding of the model, especially for the decoder.*
> Thank you for the helpful suggestion. If the paper is accepted, we’ll use the extra page available to move several details from the appendix to the main paper (including the architecture and Table 7).
>
> ### *One minor suggestion is to include the spatial pooling between the two transformers in Figure 2, since that is an important operation in obtaining the K object latents. As drawn, it is not clear that K and IJ may be different.*
> Thanks for catching this and sorry for the confusion. We have rectified this figure (and will update the draft when allowed).
>
> ### *I would be curious how SIMONe performs on more realistic scenes than the ones shown in the paper.*
> Applying SIMONe to real-world data is certainly an application we are most excited to see. It’s worth mentioning–we attempted to design SIMONe to be scalable from the outset, and to avoid any tricks/biases which could limit its performance on new datasets (see our response to Reviewer nXbF, *“What makes the authors think that SIMONe will scale to more complex scenes without major changes in the approach?”*).
>
> We’d appreciate any further questions/suggestions/feedback you may have. Thank you for your time.

---

### Official Review · Reviewer_RGhV · 2021-07-16

**Rating:** 6
**Confidence:** 4

**Summary:**

Authors propose a structured generative model of video clips. This uses a factored latent space, with per-object latents that are global the video, and temporal latents that are not split between objects. The decoder uses a spatial GMM (as in many prior works), albeit with pixels given by per-coordinate MLPs not a CNN. The encoder uses a transformer on CNN features. Results on synthetic datasets show higher performance than one existing method (S-IODINE) on video segmentation, and higher performance than GQN and NeRF-VAE on novel view synthesis.

**Limitations And Societal Impact:**

Yes, both fine.

**Main Review:**


Model etc.:

++ The model is clearly described (albeit with fairly major aspects in the supplementary).

++ The model structure is novel -- significantly different to other works in the same area, even though the components (transformers & per-pixel MLP decoding) are well-established elsewhere. Splitting a latent space into temporal and non-temporal components is also not a new idea (see below) -- this should be emphasised better. Overall I feel the novelty is sufficient.

-- It is not clear where object motion is expected to be encoded, and why. The results (Sec.5.2) state post-hoc that encoding them in the object latents is 'correct', but from the model description, they could equally be in the temporal latents.

Related work:

-- "MoCoGAN: Decomposing Motion and Content for Video Generation" [Tukyakov, CVPR 18] is relevant (it models videos and separates latents into static content and per-frame motion) and should be discussed (note various other follow-up works also used a similar factorisation).

-- "Unsupervised object-centric video generation and decomposition in 3D" [Henderson, NeurIPS 20] is relevant (it forms an object-centric latent representation of videos, with temporal information factored) and should be discussed (and/or evaluated against). In particular, it would be expected to handle the case of Fig.8 correctly, as would other 3D-aware generative models.

Experiments:

++ Pixel reconstruction of scenes is very accurate, and camera motion is successfully encoded into (only) the per-frame latents.

++ View-synthesis results are significantly better than baselines.

-- Segmentation results are poor, both quantitatively and qualitatively.

-- The model uses a novel encoder and decoder architectures (per-pixel MLP and transformer), but there is no ablation study, nor attempt to match parameter counts / architectures for the baselines. This means it is difficult to gain useful insights from the paper about what is important (e.g. maybe S-IODINE with a transformer encoder would be better? how much does the MLP decoder help vs. spatial broadcast CNN? how well does the same model work if the latent space is not factored?).

-- Evaluation is only on simple synthetic datasets where a color-based segmentation (rgb clustering with smoothness) would likely work well. Other somewhat-similar works (e.g. RELATE and O3V) use more challenging data; it would be interesting to see how SIMONe performs on their datasets.

-- Please discuss why qualitative GQN results look much worse than the original work.

---

## Post-rebuttal

The authors' response addresses several of my concerns, and I therefore am now mildly in favor of acceptance, provided missing discussion, references, etc. are added in the camera ready. Issues raised by other reviewers also seem to be adequately addressed. Note the authors' discussion in the responses here is more insightful than the original paper in some cases -- it would be good to incorporate as much as possible in the final version. The biggest remaining issues in my view are the simplistic nature of the datasets used in evaluation, and disappointing segmentation performance on them.

**Time Spent Reviewing:**

1.5

---

> ### Author Response · Authors · 2021-08-06
> **Partial response for early discussion**
>
> Thanks for your review and suggestions! We wanted to address some of your major points first to allow time for discussion. Hopefully that will clear up any confusion before we respond to the review in full:
>
> ### *Summary: Authors propose a structured generative model of video clips*
> Note that SIMONe is primarily a representation learning method, and our innovations lie largely on the inference front as well as the qualities of the learnt representations. SIMONe is not a generative model of video clips per se because it cannot generate novel clips. It is only trained via generation (i.e. reconstruction) as this provides an unsupervised learning signal.
>
> ### *The decoder uses a spatial GMM (as in many prior works), albeit with pixels given by per-coordinate MLPs not a CNN*
> SIMONe’s decoder is actually a single MLP (rather than “per-pixel” or “per-coordinate” MLPs). It is equivalent to a CNN with a 1x1 kernel. It can be thought of as inspired by/similar to a NeRF decoder.
>
> ### *There is no attempt to match parameter counts / architectures for the baselines*
> - When sweeping hyperparameters for the baseline models (MONet, S-IODINE, Slot Attention), we matched the “Large” decoder setting in each case to SIMONe’s decoder (in terms of number of layers, channels, and kernel sizes). This was critical to allow a fair comparison. Note that SIMONe’s decoder is the most computationally intensive module. Its encoder is simpler than MONet’s and S-IODINE’s.
> - See Appendix A.4 for further details on how we picked hyperparameters to sweep for the baseline models. Where hyperparameters do not match SIMONe’s (e.g. MONet’s decoder scale is set to 0.07 rather than SIMONe’s 0.08) this was done to the advantage of the baseline models.
>
> ### *Splitting a latent space into temporal and non-temporal components is also not a new idea*
> We have reviewed some relevant literature in Appendix A.1 under “Video processing and vision”, bullet (2). Given more space (e.g. at camera-ready), we will move A.1 to the main paper.
>
> Thanks for the related work you suggested. We compare them to SIMONe below. Please note in particular the comparison to O3V, where we emphasize the lack of privileged information used by SIMONe.
>
> ### Comparison to MoCoGAN [Tukyakov, CVPR 18]
> - Being a GAN, MoCoGAN focuses on video generation not scene understanding.
> - MoCoGAN is not a multi-object system. It does not infer different latent representations for different objects. It was only demonstrated to work on single-object videos.
> - Architecturally, MoCoGAN has one motion and content latent for each frame in the video. SIMONe, on the other hand, has K object latents which don’t vary across the video. These are time-averaged. We also have T frame latents (one per frame), which are space-averaged. So our latent factorization is quite different from MoCoGAN’s: SIMONe uses a total of K+T latents whereas MoCoGAN uses 2T latents.
> - Functionally, MoCoGAN separates the motion of an object from its “content” (i.e. appearance). We are doing the opposite: we encode both motion and allocentric object properties in the same latent variable. See the last answer below (*“It is not clear where object motion is expected to be encoded…”*) for an explanation of why this might be useful.
>
> ### Comparison to O3V [Henderson, NeurIPS 20]
> - O3V relies on view supervision i.e. it assumes the camera parameters are available as conditioning for each frame in a video. In SIMONe, we used "unsupervised" to indicate that no view information or camera pose is provided to the model. It is trained using RGB videos alone. (We also experimented with an ablated version of SIMONe, called SIMONe-VS, where camera information is provided to the model. But that isn’t our main contribution. Though camera parameters are indeed available in applications such as robotics, that is far from the norm in video processing and other applications.)
> - O3V explicitly handles 3D geometry. It renders objects to an “explicit voxel or mesh representation.” On the other hand, SIMONe has no built-in knowledge of 3D (such as specialized renderers), but still demonstrates a handle on 3D (e.g. see our results predicting allocentric positions for moving objects in Table 7).
>
> ### *Segmentation results are poor, both quantitatively and qualitatively*
> - Please note the lack of privileged information used by SIMONe. It does not rely on camera parameters, depth information, or proprioception and is fully unsupervised (i.e. it only observes RGB).
> - Compared to specialized computer vision or supervised learning methods, our segmentation results are indeed more limited. It is worth mentioning that segmentation is a byproduct of SIMONe’s representations, not the only purpose. Primarily, SIMONe’s representations are useful because they are (a) multi-object (b) view-invariant (i.e. close to allocentric) and (c) temporally abstracted (i.e. they summarize object motion). In our view, SIMONe’s representations achieve a useful set of desiderata for downstream tasks.
> - We do acknowledge our segmentations on Playroom are imperfect, as a Video ARI-F score of 0.8 suggests. The fact that Playroom is hard to segment should make the dataset valuable for more research when open-sourced. On the other hand, SIMONe’s segmentations on CATER reveal a genuinely accurate grasp of objects. In Figure 4, you’ll see SIMONe’s segmentation masks on CATER look rather unshapely, protruding in different directions from the core shape (whereas the ground-truth shapes are crisp). We couldn’t explain this effect until we realized there are multiple shadows per object, as the scene contains three sources of lighting. The shadows were non-trivial to identify by eyeballing the input images, but once we realized SIMONe was correct, we realized its segmentations were more useful than the ground-truth segmentations (where shadows are turned off). This effect shows that SIMONe’s spatio-temporal attention works well for segmentation, especially when coupled with object motion in the data.
> - Please also see our answer to Reviewer nXbF’s question on why our segmentations may lack sharpness (*“SIMONe [segmentations] look a lot "blobbier"...”*). One major reason is that the segmentations are decoded from latents which are subject to a KL pressure. Hence the segmentations are not comparable in sharpness to methods which don’t employ a bottleneck. (The advantage of a bottleneck is that we can uncover highly salient object features.)
>
> ### *Evaluation is only on simple synthetic datasets where a color-based segmentation (rgb clustering with smoothness) would likely work well*
> - SIMONe’s performance (in particular on Playroom) is far from “clustering by color.”  Consider the failure mode (see Animation 3 on our webpage) that SIMONe groups distant doors and windows in one slot. They are grouped together as “background” objects regardless of their distinct color, pointing at a different regularity that SIMONe uncovers.
> - On the flip side, there are several examples where SIMONe segments objects of identical color correctly. As far as we can tell, color doesn’t determine objectness as much as regularities across the dataset, composability, factors like object motion, etc.
> - To allay this concern further, we could add the segmentation scores of a color-based segmentation algorithm to Table 1 and share them here. Would that help?
>
> ### *Camera motion is successfully encoded into (only) the per-frame latents… [but] it is not clear where object motion is expected to be encoded, and why. The results (Sec.5.2) state post-hoc that encoding them in the object latents is 'correct', but from the model description, they could equally be in the temporal latents.*
> - As you've noted, the “temporal latents” are not per object. There is only one *space-averaged* temporal latent per frame. Hence, those are ideally suited to capture global properties of the scene (such as the pose of a moving observer, which affects all objects in a consistent way), rather than the independent motion of different objects.
> - Encoding object motion in the corresponding object latent is “correct” insofar as it is valuable, and in line with what we hoped for when designing our latent factorization. The value of encoding object motion *atemporally* is that a single object latent summarizes all properties of an object (including its trajectory). On a downstream task, this could allow agents to readily identify the “green flying ball” (among a set of balls represented as SIMONe object latents) or predict where that green ball would land.
> - We provide quantitative evidence that object motion is successfully encoded in the corresponding object latent alone in Table 7. Taken together, Table 2 and Table 7 present substantial evidence of the separation of information between the object and frame latents (as confirmed by Reviewer 5tHk). The two tables confirm that SIMONe can estimate the state of dynamics objects and simultaneously estimate the observer’s pose. The significance of this is described further in our response to Reviewer nXbF (*“What does it mean for an object trajectory to be captured in a "time-invariant code?"*)
>
> We hope this helps clarify these specific concerns. We will respond to your remaining questions (including our architectural choices) in detail very shortly. In the meantime, please let us know if you have any follow-ups for us.

---

> ### Author Response · Authors · 2021-08-11
> **Response #2**
>
> In this response, we address the remaining points raised by the reviewer. We continue to discuss our model choices, and comparisons to baselines/related work.
>
> ### *Please discuss why qualitative GQN results look much worse than the original work*
> We were also curious about the GQN results and looked into them extensively. There are two reasons for the poor performance:
> - One difference in the way we trained GQN (relative to the original) is we used GECO to achieve a target reconstruction score. Despite varying the target, the GECO threshold was rarely reached on Playroom data (likely due to the limitations of the unfactorized latent space–see point below). As a result, GECO lowered the weight on the GQN’s KL loss significantly, driving it to overfit to specific views. This is indeed what we see while synthesizing views around the room–GQN hallucinates objects it observes (with roughly the right shapes and colors) across novel views even when they shouldn’t be there.
> - The Playroom dataset is significantly more complex than GQN’s original Objects Room. Combinatorially, there are many objects per scene, with many possible object types and colors; this is hard to capture with a non-slotted model like GQN. Playroom also has complex relationships between objects (e.g. objects lying on furniture or the floor or each other) and an asymmetrical room. Thirdly, Playroom data is collected via the policy of an embodied agent (rather than a programmed camera). All of these contributed to GQN’s worse performance.
>
> ### *it is difficult to gain useful insights from the paper about what is important (e.g. maybe S-IODINE with a transformer encoder would be better? how much does the MLP decoder help vs. spatial broadcast CNN? how well does the same model work if the latent space is not factored?*
> - **IODINE/S-IODINE** is based on the principle that objects are refined in parallel. That is the key pressure for separation of objects (together with randomly sampled initial states to break symmetry). Our best guess is that a Transformer would not be able to use the signal from each object’s pixel likelihood to improve the representation of that particular object (a process called “refinement”); instead, it would mix influence across objects. For this reason, a Transformer-based approach has not been tried despite many follow-up publications after IODINE.
> - **SIMONe’s MLP decoder** is not very different from a spatial broadcast CNN. The key difference is that we “multisample” object latents across space or frame latents across time, whereas a spatial broadcast CNN takes a single latent sample and broadcasts it across space/time. We can certainly provide an ablation for this if it helps? We don’t believe multisampling is an essential ingredient. But we believed it could help mitigate gradient noise, which our combination of object and frame latents might be susceptible to due to the combined latent dimensionality.
> - **If the latent space is not factored**, we would obtain a latent structure close to SIMONe-VS, an ~ablation we explored thoroughly in the paper. Specifically, SIMONe-VS infers only object latents. In lieu of the frame latents (which are no longer inferred), the camera pose for each frame is provided explicitly to the model. As a result, SIMONe-VS can synthesize views from arbitrary camera positions once it derives a (multi-object) scene representation from a given context. Our results with SIMONe-VS reveal that our conception of time-invariant object latents can also work on its own (provided camera motion is explicitly given to the model), and provide more evidence that our latent factorization works as intended. We have not assessed how a view-unsupervised version of SIMONe-VS would behave, but that could also be worth exploring.
>
> ### *Other somewhat-similar works (e.g. RELATE and O3V) use more challenging data*
> - We’d be happy to try running SIMONe on one of the suggested datasets in the time remaining for rebuttal.
> - See also our response to Reviewer nXbF on why the Playroom dataset is complex enough to be challenging despite being synthetic and procedurally generated (*“Compared to supervised object instance segmentation methods, the object discovery methods work only on datasets and scenes of far lower complexity”*). In short–it’s due to the combinatorial complexity and rich choice of objects in the Playroom–features that ShapeStacks [1], TRAFFIC [2], etc fall short on.
> - See also our response on why we expect SIMONe to be scalable to more complex scenes (Reviewer nXbF, *“What makes the authors think that SIMONe will scale to more complex scenes without major changes in the approach?”*). In short, we believe it’s because of the generality of our architecture.
>
> If it’s useful, we’d be happy to share some additional analyses as suggested if the reviewer could help us prioritize which ones to run. We could try to run one or two of the following in the time remaining–(a) ablation experiments (b) results from a “color-based segmentation algorithm” on our datasets, or (c) results on one of the additional datasets you suggested. Please let us know what would be useful, and if you have any follow-up questions.
>
> Thanks again for your careful review.
>
> References:
>
> [1] Groth, O., Fuchs, F. B., Posner, I., & Vedaldi, A. (2018). Shapestacks: Learning vision-based physical intuition for generalised object stacking. In Proceedings of the European Conference on Computer Vision (ECCV) (pp. 702-717)
>
> [2] Henderson, P., & Lampert, C. H. (2020). Unsupervised object-centric video generation and decomposition in 3D. arXiv preprint arXiv:2007.06705

---

### Official Review · Reviewer_nXbF · 2021-07-17

**Rating:** 6
**Confidence:** 4

**Summary:**

The authors devise a neural network architecture, SIMONe, for learning unsupervised representations of scenes viewed from multiple camera poses. The latent representation is factored into slots meant to represent time-invariant entities (i.e., physical objects) and time-varying "frame latents," which can capture changes in the camera pose. They show that this method is able to capture the structure of scenes from three synthetic datasets by testing it on two tasks: novel view synthesis and (unsupervised) object instance segmentation. A number of qualitative results suggest that the learned latent representation successfully factors into time-invariant objects and time-varying viewpoint parameters.

**Ethical Concerns:**

No, I don't have concerns.

**Limitations And Societal Impact:**

Yes, although I think the more relevant limitations are the ones I discussed above (complex objects and learning from object motion) rather than an inability to generate completely novel scenes, for instance.

**Main Review:**

This work fits in the rapidly growing clade of unsupervised scene decomposition methods (also called "object discovery"), specifically those which try to factor scenes into K object slots plus background: MONet, IODINE, GENESIS, Slot-Attention, SPACE, SPAIR, OP3, and plenty of others. These methods are all meant to address a critical problem in AI and computer vision: how to infer and represent the structure of a scene when supervision isn't available. It's widely hypothesized (though not generally confirmed) that such object-centric representations, if they were to be used on downstream tasks, would have stronger generalization properties than unstructured representations. They're also better motivated from the standpoint of human psychology and behavior: most of our attention and goals are directed at _objects_, which can be picked out and referenced independently of the rest of a scene.

Despite this area of research getting a lot of due attention, it seems like there's still a long way to go before any of these methods will be useful for dealing with real, complex scenes. Compared to **supervised** object instance segmentation methods, the object discovery methods work only on datasets and scenes of far lower complexity (and are usually only benchmarked on simple synthetic datasets.) It's been known at least since IODINE was proposed, for instance, that the unsupervised methods fail to identify objects with multiple complex textures and shapes rather than monochromatic spheres, cones, cylinders, and so forth.

To me, the present work mostly addresses a more peripheral deficiency in the previous object discovery methods, rather than tackling the core problems with them head on. The desiderata listed at the end of the Introduction are all surely nice to have, but I think most people who work on robotics, for instance, would rather have an unsupervised object discovery method that works on real videos and objects than one that can handle a sequence of camera poses.

The authors may have a different view of what's most important or interesting to address, and below I'll try to evaluate the work based on what I think the authors were aiming for. But I think it's important for them to argue more clearly (in the Introduction and Related Work) for taking the approach that they take and adding the enhancements to prior models that they add -- in particular, because other recent works **do** seem to aim more obviously toward the central problems with unsupervised object discovery, recognize what makes them difficult, and design their methods accordingly.

_What makes unsupervised scene decomposition hard, and why doesn't this work try to address it?_

I think it's mainly two things. First, realistic objects have complicated shapes, textures, and multi-part substructures. For this reason, methods that tend to segment objects as things with uniform color or texture or by their simple shape seem destined to badly underfit on realistic scenes and datasets. The present work doesn't really expose this challenge, because even the hardest of the datasets ("Playroom") has mostly monochromatic objects with simple shapes. It's not clear that SIMONe captures even these objects especially well (see below.) So I'm not sure how much of an advance this method represents; looking at more complicated and naturally textured objects (if they can be generated in Playroom) would be one way to show this more clearly.

The second challenging thing about object discovery is that reliable learning signals are hard to come by. Just from looking at a static scene, it's impossible to know for sure what its physical structure is -- i.e., which components ("objects") can move or exist independently of the rest of the scene. Developmental psychologists have long recognized that _object motion_ is a critical source of information about how people decompose scenes into objects, and may be the main cue through which infants _learn_ how to identify physical objects in static scenes [A]. Several recent unsupervised approaches have taken inspiration from this robust literature and prioritized object motion for learning to segment scenes [B,C,D]. This has led to some success on identifying objects in more realistic (or even real) scenes; it's therefore surprising that the authors don't really mention object motion as a critical learning cue or comment on others' progress in using it. Notably, object motion could allow for learning to segment objects without simple, uniform colors and textures, since complex parts could nevertheless be grouped by "Common Fate."

So even if the authors don't think the present work is the appropriate place to solve these issues, I think the work really should at least bring them up and engage more with the literature here. If MONet, IODINE, and the like didn't already exist, there would be some logic in just developing a general approach; but at this point it's critical to identify and speak to the problems that are holding the field back.

_Specific questions and concerns_

1. The main quantitative evaluation presented is on unsupervised instance segmentation (Figure 4 and Table 1.) From what's shown, I don't quite follow how the authors are concluding that SIMONe sets a new "state of the art" on this task. Can SIMONe be tested on Static-ARI-F for direct comparison to MONet, etc. on the task for which the baselines were designed? I can see why the baselines wouldn't do as well on the Video ARI-F metric, since they're not explicitly designed to handle videos (except for S-IODINE -- but why create this "ablation" of OP3 rather than use the model that its authors intended?) Furthermore, the segmentations in Figure 4 from SIMONe look a lot "blobbier" than the ones from MONet shown in the Supplement; can the authors explain why SIMONe nevertheless has higher scores on this dataset?

2. The qualitative analysis of the learned representations is a little cryptic. For example: "The rubber duck does not morph into a chest of drawers because those are always located against a wall. This indicates a well-structured latent representation, which the decoder can adapt to" (L236-241). Why does this indicate a well-structured latent? In other methods, would one expect the rubber duck to morph into a chest of drawers?

3. I didn't really understand section 5.2 ("Temporal abstraction"). The authors seem to be making a point about how, in scenes where one or more object is moving, this motion is captured in the object latent rather than the frame latents, and that object motion is consistent across viewpoints. But what does it mean for an object trajectory to be captured in a "time-invariant code?" Is the idea that the interaction (in the Transformer modules) between the time-averaged object latents and the space-averaged frame latents can give rise to object motion?

4. The authors point out that their architecture doesn't have an explicit dynamical model (e.g. a recurrent GNN) like OP3 or some others. But that seemed liked a strength of other models -- they could extrapolate object positions into the future (a little.) Would it be possible to do that with SIMONe after some modification?

5. Related to my higher-level concerns above, the section on "Scalability" seems a little misdirected. What makes the authors think that SIMONe will scale to more complex scenes without major changes in the approach?

6. Without at least some ablation studies, it's hard to know how important particular design choices are. Is the Transformer contributing significantly to this architecture, or could other latent state inference modules (like a recurrent GNN) serve just as well?


[A] Spelke, Elizabeth S. "Principles of object perception." Cognitive science 14.1 (1990): 29-56.

[B] Veerapaneni, Rishi, et al. "Entity abstraction in visual model-based reinforcement learning." Conference on Robot Learning. PMLR, 2020.

[C] Bear, Daniel M., et al. "Learning physical graph representations from visual scenes." arXiv preprint arXiv:2006.12373 (2020).

[D] Du, Yilun, et al. "Unsupervised Discovery of 3D Physical Objects from Video." arXiv preprint arXiv:2007.12348 (2020).



**Time Spent Reviewing:**

5

---

> ### Author Response · Authors · 2021-08-06
> **Partial response for early discussion**
>
> Thanks for your thoughtful and detailed review, suggestions about the paper’s framing, and specific questions. We wanted to address some of your major points first to allow time for early discussion. Hopefully that will clear up any confusion before we respond to the review in full:
>
> ### *What does it mean for an object trajectory to be captured in a "time-invariant code?"*
> - Thanks for raising this – it’s rather central to our work and worth clarifying in detail. Consider a simple example of an object thrown into air with some initial velocity. A model could either encode its position at every time point; or assuming it understands gravity (say via learning), it can summarize the object’s trajectory using its initial velocity and a method to “query” (i.e. compute) its position at any given time. The latter is more economical in terms of information required, and we’d refer to the initial conditions/velocity of the object as the “time-invariant code” for this particular example. They serve as sufficient statistics.
> - SIMONe is capable of summarizing objects’ trajectories in their time-invariant object latents $\mathbf{o}_k$. They are “time-invariant” by construction of our latent factorization (as they are fixed across the whole sequence); these object latents can then be queried together with time using a function $f(\mathbf{o}_k, t)$ to obtain object k’s trajectory. We provide quantitative evidence of this in Table 7, where we examine how accurately an object’s trajectory is encoded in SIMONe’s corresponding object latent on the CATER dataset. We intend to move this table to the main paper in addition to Table 2.
> - Note that given our factorization, one could imagine an alternative solution where the model encodes the positions of all objects in the frame latent at every time point (thus using the object latents for non-dynamical information only). We found this not to be the case, almost certainly because it is a less economical solution under the capacity constraints enforced by our KL regularizers. It would require storing all objects' positions in a single frame latent at every step, disregarding physical regularities over time; and it wouldn’t take advantage of the causal independence of objects in a scene.
> - So why is it useful for an object trajectory to be captured in a “time-invariant code”? Say a system needs to classify rolling balls versus static ones. Or flying balls which will go into a hoop versus ones that will miss. Given access to each SIMONe object latent, the system has all the information it needs to make those decisions.
>
> ### *SIMONE doesn't have an explicit dynamical model like OP3 or some others. But that seemed liked a strength of other models -- they could extrapolate object positions into the future (a little.) Would it be possible to do that with SIMONe after some modification?*
> - We showed in Table 7 that SIMONe’s object representations can be used to query object positions for time-steps $1 \le t \le T$. This could in fact be extended to future time-steps with **no** modification of the architecture. All we need to do is query the object representation using the same trajectory decoder $f(\mathbf{o}_k, t)$, but with $t > T$. Inasmuch as an object latent captures the initial conditions/summary statistics of the object’s trajectory, the trajectory decoder (or pixel decoder) should be able to extrapolate the trajectory beyond the observed data. While we haven’t tested this extrapolation, nothing in the model prevents it. We’d be happy to add such an experiment in the final manuscript.
> - Alternatively, as you suggest and we alluded to in the paper, one could explore a hybrid approach with an explicit dynamics model. We haven’t had the opportunity to do that yet.
>
> ### *SIMONe [segmentations] look a lot "blobbier" than the ones from MONet shown in the Supplement; can the authors explain why SIMONe nevertheless has higher scores on this dataset.*
> - Regarding higher ARI scores: blobbiness is not penalized if a blobby mask overlaps with the background. The ARI-F score only considers foreground pixels. We use ARI-F instead of a background-inclusive ARI (more or less standard in related work) because the latter obfuscates segmentation quality due to the weight of the background.
> - Regarding blobbier masks than MONet: the blobbiness could be an artefact of camera motion. SIMONe captures the position and extent of an object across multiple timesteps. So blobbiness could reflect some uncertainty about the exact location of an object at a particular timestep. MONet on the other hand only processes one frame at a time. In addition, its pre-bottleneck attention network (a U-Net with skip connections) can extract arbitrarily precise segmentations, whereas SIMONE’s post-bottleneck (decoded) segmentations are constrained by the information bottleneck in the latents.
> - So far as the blobbiness of masks doesn’t lead to undersegmentation/missed objects, and small or intricate objects are handled reasonably, we don’t consider this a huge issue for downstream tasks. (Representing an object’s shape with pixel-level precision is more expensive and probably less important for downstream tasks than capturing its allocentric position, for instance).
> - Please also see our response to Reviewer RGhV’s comments regarding SIMONe’s segmentation quality more generally (*“Segmentation results are poor, both quantitatively and qualitatively.”*)
>
> ### *Can SIMONe be tested on Static-ARI-F for direct comparison to MONet, etc. on the task for which the baselines were designed?*
> - Video ARI-F generally serves as a lower bound for the Static ARI-F. (Video ARI treats segments spatio-temporally – like "tubes" through the space time volume. This penalizes models which switch slots between time-steps as it breaks consistency across the tubes). SIMONe’s Video ARI-F performance alone is competitive to the baselines’ Static-ARI-F performance. To satisfy the reviewer’s concern, we could provide a curve showing SIMONe’s Static ARI-F score per time step if that helps?
> - MONet was originally designed and tested on a version of Objects Room. We included Objects Room in our choice of datasets precisely to allow a direct comparison with SIMONe.
>
> ### *I think most people who work on robotics, for instance, would rather have an unsupervised object discovery method that works on real videos and objects than one that can handle a sequence of camera poses*
> - Our method goes beyond handling a sequence of camera poses. Given its ability to represent allocentric object positions, it effectively constructs an allocentric object-centric map of the scene, estimating the state of dynamic objects (see Table 7), and simultaneously estimating the observer’s pose (see Table 2). As discussed in our Related Work, this amounts to taking baby steps toward unsupervised object-centric SLAM, a problem of large-scale significance in robotics.
> - Moreover, handling changing camera pose correctly without explicit view supervision is crucial for applications to real videos (where camera pose, unlike in robotics, is not readily available).
> - See also our response below to why we expect SIMONe to scale to more complex and real world scenes.
>
> ### *What makes the authors think that SIMONe will scale to more complex scenes without major changes in the approach?*
> It’s mainly the choice and generality of our architecture. A transformer-based encoder coupled with a decoder that queries representations by space and time seem readily scalable for the following reasons:
> - **Inductive bias**: joint spatio-temporal attention ensures we maximally utilize the training signal across space and time. It seems like a natural choice to allow interaction between many “things.” Given that the transformer attention mechanism uses dot product similarity to weigh interactions between slots makes it straightforward to identify multiple occurrences of an object over time.
> - **Computational complexity**: SIMONe can process (i.e. extract) objects in a full sequence in a parallel manner (even the initial CNN can be applied in parallel across frames). It can also extract more objects per frame than several existing methods (notable exception: SPACE). The low computational complexity of the encoder and decoder (compared to autoregressive encoders/decoders) implies we can ultimately process more data/frames.
> - **The field at large**:
>   - Vision transformers (ViT), DINO [1], etc are making strides on real world data in a similar fashion to SIMONe (we’ve compared them to SIMONe in Appendix A.1 under “Video processing and vision”). For the decoder, NeRF-like models have shown tremendous success in querying representations by viewpoint.
>   - Research on transformers is moving at lightning speed. Any improvements made in the field should translate to scalability benefits for SIMONe, given that we used the simplest possible transformer so far.
>   - As speculated in the paper, we could borrow techniques from the use of transformers in language models to process long sequences (perhaps using “memory slots” and applying SIMONe in a sliding window).
>
> For the reasons above, we expect SIMONe will be scalable in the number of objects inferred, number of frames processed, as well as number of pixels decoded, making it highly amenable for future research.
>
> We hope this helps clarify these specific concerns. We will respond to your remaining questions (encompassing textured objects, object motion, and comparisons to PSGNet and POD-Net) in detail very shortly. In the meantime, please let us know if you have any follow-ups for us.
>
> References:
>
> [1] Caron, M., Touvron, H., Misra, I., Jégou, H., Mairal, J., Bojanowski, P., & Joulin, A. (2021). Emerging properties in self-supervised vision transformers. arXiv preprint arXiv:2104.14294.

---

> ### Author Response · Authors · 2021-08-11
> **Response #2**
>
> In this response, we continue to address the possible misunderstanding that SIMONe segments/represents *“time-invariant objects”* (plus “time-varying viewpoint parameters”) while failing to use *“object motion”* for object identification. This led the reviewer to conclude that SIMONe might *“segment objects as things with uniform color or texture, or by their simple shape,”* a tendency which would be *“destined to badly underfit on realistic scenes and datasets.”*
>
> 1. Note firstly that SIMONe doesn’t work exclusively with “time-invariant objects.” It does **handle dynamic objects** by summarizing their trajectories invariantly of time. Perhaps our choice of terminology (“time-invariant object latents”) has been the cause of some confusion. See our previous response for a description of how we’re using the phrase “time-invariant.”
> 2. Note also that SIMONe isn’t applied frame by frame. It **processes a whole sequence** (of consecutive frames) with one pass through the encoder. Hence motion is an important cue for segmentation (along with regularity across the dataset and diverse composition of scene elements.)
> 3. While object motion is certainly a valuable cue for segmentation (including in infants), **camera motion is** arguably **also fundamental** (providing cues for segmentation via motion parallax, etc). It is not sufficient to rely on object motion alone (e.g. a child might learn to recognize a bed as a distinct object long before he/she sees a bed move). Hence our contributions are on both fronts.
> 4. On the **qualitative differences between SIMONe’s segmentations versus prior work**:
>     - Consider **MONet**’s segmentations on Playroom data (Figure 16). MONet consistently merges the brown base of each bed with the brown floor, thus exhibiting a tendency for clustering by color. It also groups toys/small objects by color (e.g. the blue mattress and blue-windowed bus in column 4 or the purple duck and purple cushion in column 5 of Figure 16). On the other hand, SIMONe segments the full (two-colored) bed as one object consistently, and we don’t see it group the aforementioned toys by color.
>     - When **Slot Attention** segments CATER, it ignores object shadows completely. It infers crisp shapes, showing a clear propensity to use each object’s uniform color and lack of texture. SIMONe on the other hand assigns every object’s shadows (up to three) in the corresponding object’s segment.
>     - Our method is completely unrelated to **IODINE/S-IODINE** and there’s no reason to expect it will discover comparable “objects” or have the same deficiencies. IODINE/S-IODINE can barely handle more than 7 or 8 objects. Given SIMONe’s K=16 object latents, that alone is an important reason to expect different failure modes than S-IODINE.
> 5. SIMONe’s segmentation capabilities through time are not evident from the static figures in the paper. Please see our animated figures. SIMOne’s **stable tracking of objects** (on both CATER and Playroom) demonstrates a clear understanding of object and camera motion.
>
> ### *It's therefore surprising that the authors don't really mention object motion as a critical learning cue*
> - We share the reviewer’s view that object motion is a powerful signal for object discovery and multi-object scene representation. That is precisely why SIMONe works on videos (rather than static images). Our results on CATER show that SIMONe accurately captures object motion (Table 7), which certainly contributes to its segmentation performance on that dataset.
> - We will further emphasize the importance of object motion as a learning signal in our revision. We’ve already covered some work that uses object motion in our Related Work and Extended Related Work (see Appendix A.1) but will also add and highlight the papers pointed out by the reviewer.
>
> ### *It's therefore surprising that the authors don't… comment on others' progress in using [object motion]*
> #### Comparison to POD-Net
> - POD-Net makes a valuable contribution in showing how object discovery methods can be extended to large images containing objects of different sizes. Attending to sub-patches of an image seems like a general approach (given the relation to foveated vision) that many models (including SIMONe) could use and benefit from. That said, the approach doesn’t have an easy extension to videos. Merging segments across time based on pixel overlap wouldn’t work at all when object motion or camera motion is substantial.
> - POD-Net’s dynamics approach is also limited to uniform motion (it simply averages historical velocity to predict motion, which implies it can’t cope with simple cases like objects accelerating in free fall).
> - The use of supervised backprojection and projection modules, which are trained on single-object data, also gives POD-Net a leg up. Models like SIMONe don’t get any prior knowledge about what constitutes a single object (or how to infer its size/allocentric position from a 2D image). SIMONe’s notion of objectness and handling of 3D is *emergent* and *implicitly derived*. (This is a recurring theme in subsequent answers.)
>
> #### Comparison to PSGNet
> - We will certainly update our paper to acknowledge PSGNet.
> - We do think PSGNet is a great example of a method that explicitly uses object motion (through self-supervision losses inspired by human perception). In contrast, SIMONe uses object motion *implicitly*, and in conjunction with other factors. SIMONe’s CATER results (see Table 7 and Animation 3 on our webpage showing the stable tracking of objects along with their shadows) do affirm that SIMONe makes use of object motion when present in the data. While explicit (object motion-based) losses weren’t required in our case so far, exploring when this is or isn’t the case would be valuable future work.
> - The use of depth maps and normal maps makes PSGNets infeasible to apply broadly. With SIMONe, we were interested in pushing the limits of unsupervised learning (from RGB data) without auxiliary inputs or engineered loss terms to maximize possible use cases. In our view, a model that achieves an *emergent understanding* of object motion from RGB visual data is interesting on its own.
>
> ### *Compared to supervised object instance segmentation methods, the object discovery methods work only on datasets and scenes of far lower complexity (and are usually only benchmarked on simple synthetic datasets.) It's been known at least since IODINE was proposed, for instance, that the unsupervised methods fail to identify objects with multiple complex textures and shapes rather than monochromatic spheres, cones, cylinders, and so forth*
> - The comparison to supervised instance segmentation methods is not strictly appropriate, as we’re explicitly tackling an unsupervised version of the problem. If research on ImageNet stopped after the success of supervised methods, we would never know what is possible via an unsupervised approach (e.g. SimCLR [1], SCAN [2]).
> - While it is a synthetic and structured environment, the Playroom was designed to go beyond the simplicity of monochromatic, convex objects. Consider the bookcase, table, or stool for examples of intricate shapes. Consider the windows, beds, chests of drawers, soccer and basketballs, rubber ducks, or even the agent’s avatar for examples of multi-colored objects. Consider the window or soccer ball for examples of texture. The fact that unsupervised segmentation on Playroom is yet to be “solved”–and leads to substantially different qualitative results–by the variety of baselines we tested is testament to its challenge.
> - Given our use of camera motion and object motion and joint spatiotemporal inference, we believe SIMONe exhibits completely different failure modes than IODINE or MONet. See our response above (“On the qualitative differences between SIMONe’s segmentations versus prior work”) for more details.
>
> ### *Why create this "ablation" of OP3*
> S-IODINE is described with results in Appendix A.1 of Greff et al 2019 (IODINE). Only the “S-” prefix is of our creating (to avoid confusion). S-IODINE seemed more appropriate to use on our sequential data rather than the basic version of IODINE. While it can be considered an ablation of OP3, that was not our ultimate intention.
>
> ### *Without at least some ablation studies, it's hard to know how important particular design choices are. Is the Transformer contributing significantly to this architecture, or could other latent state inference modules (like a recurrent GNN) serve just as well?*
> - Our original choice of Transformers was motivated by their ability to capture generic interaction between many components, which was desirable to handle objects across space and time alike. We also appreciated them for their scalability (as discussed in the previous response). They suffer less overhead than autoregressive attention- or refinement-based inference modules (e.g. MONet’s U-Net or IODINE’s gradient-based refinement).
> - Our results and claims are centered on what can be achieved using our chosen architecture. (The fact that joint spatio-temporal attention can be used effectively to extract objects from sequences has not been previously explored.) We don’t claim that the Transformer architecture is necessarily better than any alternatives, but merely that it can do XYZ. For that reason, comparing against Transformer alternatives wouldn’t be an “ablation” study as it wouldn’t help reinforce any of our actual claims.
> - Future work could certainly try using alternatives such as recurrent GNNs which also allow interaction between objects. We do expect SIMONe will be amenable to different choices of architecture or alternative/improved implementations.
>
> Apologies for not addressing your questions pointwise, but we’ve covered most of the topics raised. Do please let us know if you’d also like a pointwise response. And thanks once again for a very detailed review, raising both the technical merits of our work and its wider significance in the field.

---

> > ### Author Response · Authors · 2021-08-11
> > **References for response #2**
> >
> > [1] Chen, T., Kornblith, S., Norouzi, M. &amp; Hinton, G.. (2020). A Simple Framework for Contrastive Learning of Visual Representations. Proceedings of the 37th International Conference on Machine Learning, in Proceedings of Machine Learning Research 119:1597-1607 Available from http://proceedings.mlr.press/v119/chen20j.html.
> >
> > [2] Van Gansbeke, W., Vandenhende, S., Georgoulis, S., Proesmans, M., & Van Gool, L. (2020, August). Scan: Learning to classify images without labels. In European Conference on Computer Vision (pp. 268-285). Springer, Cham.

---

> > > ### Comment · Reviewer_nXbF · 2021-08-30
> > > **Thank you and sorry for the delay!**
> > >
> > > Thank you for your very thorough response to my concerns. Since of them had to do with the framing of the paper, comparison to other recent progress, and a poor understanding of some of your claims, I think your paper will be much stronger if you can use the extra space provided in revision to include some of this discussion. I will raise my score and support acceptance of this paper.
> > >
> > > (I wrote much more detailed responses to your comments but sadly they got lost in the internet aether. I'll try to rewrite some of them in further comments below, but want to make sure my score change takes effect first.)

---

> > > > ### Comment · Reviewer_nXbF · 2021-08-30
> > > > **Static ARI and color-based clustering method**
> > > >
> > > > In your response to me and reviewer RGhV, you raised the possibility of running Static-ARI on SIMONe and trying a simple baseline based on color segmentation, respectively. I think these are important controls and suggest including them in the final manuscript.

---

> ### Author Response · Authors · 2021-08-20
> **Response #3**
>
> Apologies, we missed two points you raised and are addressing them below:
>
> ### *To me, the present work mostly addresses a more peripheral deficiency in the previous object discovery methods, rather than tackling the core problems with them head on.*
> SIMONe tackles three core deficiencies in existing methods. Here are brief reasons why we believe these deficiencies are critical and why our approach is worthwhile:
> - **Allocentric object representations**: State-of-the-art vision systems still struggle with multi-object reasoning (barring explicit supervision), even in simple environments. When presented with (for instance) a Playroom-like scene and asked “which of the green balls is larger” or “which of them is closer to the window,” there are multiple issues that existing systems need to overcome: they need to understand how the scene is composed and the objects’ characteristics. They need to overcome issues of perspective–namely that depending on the viewer’s pose, one of the green balls might appear larger regardless of their actual sizes. If one of the green balls happens to be rolling/flying, the system also needs to understand their dynamics. These variables are nontrivial to infer together.
> - **Stable object tracking**: Temporal reasoning is only possible when objects can be tracked stably. A system that uses both an object’s appearance and its dynamics to track it (as SIMONe does by summarizing both kinds of information in a single object latent) can deal correctly with more edge cases (e.g. rolling balls that are briefly eclipsed or a collision that causes position overlap) than a system that relies on only one of those features.
> - **Simultaneous estimation of camera pose and dynamic object states**: SLAM is a classic chicken and egg problem [1, 2] which we’re approaching using a novel variational approximation (i.e. factorizing our latent space into object and frame latents), via learning, and without supervision (but admittedly with other simplifications.)
>
> With SIMONe, we’re taking a stab at addressing these deficiencies at once using a fairly general method. See also our previous response about SIMONe’s relevance to robotics (*I think most people who work on robotics...*) and scalability to more complex/real world scenes (*What makes the authors think that SIMONe will scale to more complex scenes*).
>
> ### *The qualitative analysis of the learned representations is a little cryptic. For example: "The rubber duck does not morph into a chest of drawers because those are always located against a wall. This indicates a well-structured latent representation, which the decoder can adapt to" (L236-241). Why does this indicate a well-structured latent? In other methods, would one expect the rubber duck to morph into a chest of drawers?*
> - We admit this comment may be confusing–it was meant to share intuition about the model’s behavior on the Playroom dataset. It is debatable what a well-structured latent should be. In our case, it would’ve been odd if the method learnt to transform a chest of drawers into a rubber duck because that’s not a transformation supported by the data. The environment treats toys very differently to furniture when a scene is procedurally generated.
> - Regarding other methods, the key point is it is hard for them to learn object identity transformations *at all*, as they (often) lack the means to learn latents corresponding to the *allocentric* structure of the scene. We wanted to highlight SIMONe’s achievement on that front given Playroom’s non-trivial layout, object diversity, arbitrary agent trajectories, etc.
>
> Thanks again for giving us the opportunity to clarify these details.
>
>
> References:
>
> [1] http://ais.informatik.uni-freiburg.de/teaching/ss17/robotics/slides/22-summary.pdf
>
> [2] ​​https://www.cs.columbia.edu/~allen/F19/NOTES/slam_paper.pdf

---

### Official Review · Reviewer_Uh4X · 2021-07-18

**Rating:** 8
**Confidence:** 4

**Summary:**

This submission presents a video scene model that decomposes a scene into object components, and frame components in an unsupervised way. The architecture improves over previous models that are static (MONET, IODINE, etc) and use videos (GENESIS) in that the model performs better and is conceptually simpler. The main idea is the use of a transformer model that stacks all feature representation generated from CNNs for every individual frame, then aggregate to object and frame latents.

Experimental results are shown on synthetic images only. Comparison are done in two settings: knowing the camera locations (against GQN and Nerf-VAE) and fully unsupervised (compared against MONET and SlotAttention). Empirical results verify the model to be stronger.

**Limitations And Societal Impact:**

The limitations are sufficiently addressed. Negative societal impact is abstract for this work and is shared with all models that improve on scene understanding in automated ways.

**Main Review:**

Well written paper that is an extension of current work on unsupervised models that decompose scenes into its constituents. This model is novel in the way it uses transformers to decompose into frame and object latents. This makes for a conceptually simple model. Experimental results are sufficient, the model is tested in two scenarios (with and without known camera position) and performs better than reasonable chosen baselines. The submission presents some weak evidence of the disentangling of the object/frame latents in that the camera position is not recoverable from the objects but only from the frame representation.

The main drawbacks are
- This is no fully generative model. The submission comments on this in the future work and limitation section. I concur with the potential extension borrowed from IODINE.
- Experimental results are shown on synthetic images only. The Nerf-VAE baseline is a fair competitor for this model and this type of date. The question whether these scene models will generalize to more complex environments is still open. This submission improves on the technical side but not on the experimental one.

I recommend acceptance of this paper, I enjoyed reading it and the level of detail is sufficient to understand the details of the method.

Did you experiment with the inclusion of optical flow into the model? Would that not potentially improve on the attention mechanism? What are the main obstacles for using optical flow?

**Time Spent Reviewing:**

4

---

> ### Author Response · Authors · 2021-08-06
> **Acknowledgement and minor clarification**
>
> Thanks for your positive review and comprehensive comments on the evaluation, potential extensions, and limitations of our model! We're glad you appreciated the different aspects of the paper. Before responding to the review in full, we wanted to share a quick clarification and solicit relevant suggestions:
>
> ### *The submission presents some weak evidence of the disentangling of the object/frame latents in that the camera position is not recoverable from the objects but only from the frame representation.*
> - In addition to the evidence you’ve noted, we also present evidence in Table 7 (Appendix) that the position of a moving object at any time-step can be recovered from the corresponding object latent (when queried with time). This helps assert our claim that the object latents summarize object dynamics *invariantly of time* and *invariantly of camera pose*.
> - Taken together, Table 2 and Table 7 present substantial evidence of the separation of information between the object and frame latents (as also confirmed by Reviewer 5tHk). We would appreciate any further suggestions you may have on how to strengthen the evidence.
> - By estimating the state of dynamic objects and simultaneously estimating the observer’s pose, SIMONe takes baby steps toward unsupervised object-centric SLAM, a problem of large-scale significance even beyond our immediate applications toward segmentation and novel view synthesis. Together with the scalability of the model (see our response to Reviewer nXbF, *“What makes the authors think that SIMONe will scale to more complex scenes without major changes in the approach?”*), we expect SIMONe will be highly amenable to future research.
>
> Please let us know if you have any follow-ups for us (barring additional experiments as we’re not allowed to update the paper draft). We will respond to the remaining questions/suggestions in detail very shortly.

---

> ### Author Response · Authors · 2021-08-11
> **Main response**
>
> Thanks again for the time you spent studying the paper in detail. We address the points you raised below.
>
> ### *This is no fully generative model*
> Indeed, that is a limitation. Thanks for validating the potential extensions discussed in the paper as ways to overcome it.
>
> ### *Experimental results are shown on synthetic images only… This submission improves on the technical side but not on the experimental one.*
> - We certainly agree that one of the core challenges in the subfield is to raise the standard of evaluation to non-synthetic data. The main impediment to realizing this is one of the following: either lack of structure across different scenes (to allow common-format multi-object representations) or lack of ground-truth masks (to evaluate segmentation).
> - Nevertheless, we’ve attempted to push the needle on experimental evaluation as follows:
>   - Reporting both Static and Video ARI-F scores aligns us with work in video processing (where stable tracking is as critical as good segmentation).
>   - In evaluating SIMONe-VS on view synthesis, we’re using an experimental technique common in NeRF-like scene rendering models in the context of object-centric representation learning. (We just realized recently that ROOTS [1] also did this–we will certainly cite that paper in our work.)
>   - We paid attention to assessing the information content of our representations by decoding camera pose and moving object positions from our frame latents and object latents (respectively), in addition to visualizing single attribute traversals for both latent types.
> - On the dataset front:
>   - The creation and (pending) release of our Playroom dataset (designed to encompass non-trivial object types, relationships between objects, large compositional variety, erratic camera motion, etc) will provide a new competitive benchmark to be solved.
>   - We will also release CATER data with the ground-truth masks we generated.
> - We hope these contributions will help raise the standard of experiments and analysis for future work in this area.
>
> ### *Did you experiment with the inclusion of optical flow into the model? Would that not potentially improve on the attention mechanism? What are the main obstacles for using optical flow?*
> We haven’t experimented with including optical flow so far. While it is possible and likely to improve results, we felt it would take away from the unsupervised nature of the model. Furthermore, since the Transformer allows every spatial feature map to attend all others, there is the potential for SIMONe to learn an implicit form of optical flow by mapping a part of a scene across adjacent frames. Recently proposed models such as RAFT [2] or Perceiver [3] share this aspect of the model to some extent, and could even provide a template to implement optical flow estimation in SIMONe.
>
> We’d appreciate any further questions/suggestions/feedback you may have.
>
> References:
>
> [1] Chen, C., Deng, F., & Ahn, S. (2020). Learning to infer 3d object models from images. arXiv preprint arXiv:2006.06130.
>
> [2] Teed, Z., & Deng, J. (2020, August). Raft: Recurrent all-pairs field transforms for optical flow. In European conference on computer vision (pp. 402-419). Springer, Cham.
>
> [3] Jaegle, A., Borgeaud, S., Alayrac, J. B., Doersch, C., Ionescu, C., Ding, D., ... & Carreira, J. (2021). Perceiver IO: A General Architecture for Structured Inputs & Outputs. arXiv preprint arXiv:2107.14795.

---

### Author Response · Authors · 2021-08-20
**Meta-comment on key features**

We thank all reviewers for their thorough feedback. There’s a lot of great questions and suggestions/feedback on where the writing could be strengthened.

We wanted to emphasize the following key points in light of which we hope SIMONe should be judged:
1. Throughout the paper, we take “unsupervised” to mean no camera/viewpoint information is used.
2. SIMONe’s architecture, training, and potential applications are very general. Note that the system doesn’t use/need any of the following:
    - Non-RGB data (e.g. depth information, normal maps, camera poses, agent actions/proprioception).
    - Hand-crafted losses.
    - Pre-processing of data (e.g. smoothing, dividing into patches) except cropping/resizing.
    - Pre-training or sequential training of modules.
    - Specialized deep learning architectures (beyond transformers and MLPs).
    - Hyperparameters beyond the decoder scale and beta weights in the loss (hyperparameters like the number of objects K, number of layers, etc could be tuned further but need not be; they could be retained with reasonable results for any dataset).
3. The separation of information achieved by SIMONe (between the object and frame latents) is significant and useful. It happens because the object latents are time-averaged whereas the frame latents are space-averaged. It is also forced by the KL pressure compressing the latents. In summary, we found that:
    - The object latents encoded
      - allocentric position/size. This leads to the notion of “view invariant” representations.
      - shape and object identity
      - object motion. This leads to the notion of “temporally abstracted” object representations.
    - The frame latents encoded:
      - Camera pose
      - Scene lighting

We’d be happy to answer any questions and certainly welcome any suggestions to improve the paper.

---

### Decision · Program_Chairs · 2021-09-27

**Decision:**

Accept (Poster)

**Comment:**

This paper presents a method for learning disentanglement of scenes (as seen by an agent) into scene related content and object related content. It is based on transformers and a latent representation separated into time-varying and time-invariant slots.

The paper received 4 expert reviews, and in the discussion phase very quickly a consensus emerged on acceptance.

Some weaknesses were raised, in particular with experiments, positioning and mid- and long-term objectives of this line work, and the advantages or disadvantages of key design choices (missing ablations). The authors provided a thorough and detail review, which answered most questions related to framing the method, and lead to increased ratings.

All reviewers agreed that this paper is of interest to the community and recommend acceptance.
The AC concurs.